# THERE WAS NEVER A BOTTLENECK IN CONCEPT BOTTLENECK MODELS

**Antonio Almudévar**
ViVoLab, I3A
University of Zaragoza
almudevar@unizar.es

**José Miguel Hernández-Lobato**
Machine Learning Group
University of Cambdridge
jmh233@cam.ac.uk

**Alfonso Ortega**
ViVoLab, I3A
University of Zaragoza
ortega@unizar.es

## ABSTRACT

Deep learning representations are often difficult to interpret, which can hinder their deployment in sensitive applications. Concept Bottleneck Models (CBMs) have emerged as a promising approach to mitigate this issue by learning representations that support target task performance while ensuring that each component predicts a concrete concept from a predefined set. In this work, we argue that CBMs do not impose a true bottleneck: the fact that a component can predict a concept does not guarantee that it encodes only information about that concept. This shortcoming raises concerns regarding interpretability and the validity of intervention procedures. To overcome this limitation, we propose Minimal Concept Bottleneck Models (MCBMs), which incorporate an Information Bottleneck (IB) objective to constrain each representation component to retain only the information relevant to its corresponding concept. This IB is implemented via a variational regularization term added to the training loss. As a result, MCBMs yield more interpretable representations, support principled concept-level interventions, and remain consistent with probability-theoretic foundations.

## 1 INTRODUCTION

Most machine learning models operate by learning data representations—compressed versions of the input that retain the essential information needed to solve a given task (Bengio et al., 2013). However, these representations often encode information in ways that are not easily interpretable by humans. This lack of interpretability becomes especially problematic in sensitive domains such as healthcare (Ahmad et al., 2018; Xie et al., 2020; Jin et al., 2022), finance (Brigo et al., 2021; Liu et al., 2023), and autonomous driving (Kim & Canny, 2017; Xu et al., 2024b). To address this issue, *Concept Bottleneck Models* (CBMs) have been proposed, which enforce representations to be defined in terms of a set of human-understandable concepts (Koh et al., 2020).

Given an input $x$ and target $y$, Vanilla Models (VMs) are trained with a single goal: the representation $z$ derived from $x$ should encode all information needed to accurately predict $y$. In many settings, additional side information—often referred to as concepts—is available, denoted by $c = \{c_j\}_{j=1}^m$. Concept Bottleneck Models (CBMs) leverage this by extending VMs with a second objective: each concept $c_j$ must be recoverable from a designated component $z_j \in z$. By enforcing this additional constraint, CBMs are purported to provide: (i) enhanced interpretability of the learned representation space, and (ii) the ability to perform targeted interventions on specific concepts by manipulating $z_j$ and propagating the resulting changes to the model's predictions.

However, CBMs are prone to a phenomenon known as *information leakage* (Margeloiu et al., 2021; Mahinpei et al., 2021), where the representation $z$ encodes input information that cannot be attributed to the predefined concepts $c$. We refer to this additional information as nuisances $n$. Information leakage raises two main concerns: (i) it undermines interpretability, since $z_j$ cannot be fully explained by its corresponding concept $c_j$; and (ii) it compromises the validity of interventions—modifying $z_j$ may alter not only the associated concept $c_j$, but also other unintended information encoded in $z_j$.

We argue that *information leakage* stems from a fundamental limitation in the current formulation of CBMs: the absence of an explicit Information Bottleneck (IB) (Tishby et al., 2000) that actively constrains $z_j$ to exclude information unrelated to $c_j$. While the second objective in CBMs encourages each $z_j$ to retain $c_j$ in its entirety, it does not enforce that $z_j$ captures only information about $c_j$. In the worst-case scenario, $z_j$ could encode the entire input $x$ and still satisfy this objective.

To address this issue, we propose *Minimal Concept Bottleneck Models* (MCBMs), which incorporate an IB into each $z_j$. This ensures that $z_j$ not only retains all the information about its associated concept $c_j$, but also ex-

Table 1: Differences between VMs, CBMs and MCBMs.

|  | VMs | CBMs | MCBMs |
|---|---|---|---|
| Does $z_j$ encode *all* $c_j$? | ✗ | ✓ | ✓ |
| Does $z_j$ encode *only* $c_j$? | ✗ | ✗ | ✓ |

cludes any information unrelated to $c_j$, as summarized in Table 1. The name reflects that $z_j$ is trained to be a *minimal sufficient* statistic of $c_j$ (Fisher, 1922; 1935), in contrast to traditional CBMs where $z_j$ is optimized to be merely a *sufficient* statistic of $c_j$. As illustrated in Figure 1, this design yields disentangled representations that directly address the two shortcomings of CBMs: (i) it improves interpretability, since $z_j$ can be fully explained by its corresponding concept $c_j$; and (ii) it enables valid interventions—modifying $z_j$ affects only the associated concept $c_j$.

In Section 2, we connect the data generative process to MCBMs through information-theoretic quantities, showing that the IB can be implemented via a variational loss. In Section 3, we review alternative approaches to address information leakage. In Section 4, we present experiments demonstrating that MCBMs enforce a true bottleneck, thereby enhancing interpretability and intervenability compared to existing alternatives. Finally, in Section 5, we show that the assumptions made in CBMs to enable interventions are theoretically flawed.

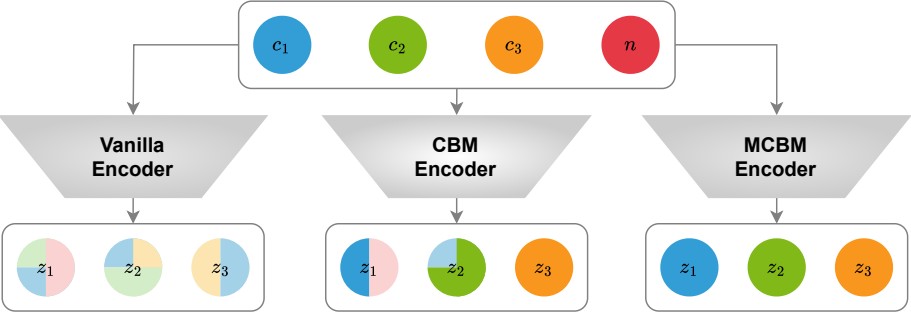

Figure 1: In Vanilla Models, concepts and nuisances may be arbitrarily entangled in the representation, and a variable $z_j$ may capture only part of a concept (depicted as paler colors). In CBMs, each $z_j$ encodes all information about its corresponding concept $c_j$, but may also capture some information about nuisances (e.g., $z_1$) or other concepts (e.g., $z_2$). In contrast, MCBMs enforce that each representation variable $z_j$ encodes all—and only—the information about its corresponding concept.

## 2 FROM DATA GENERATIVE PROCESS TO MCBMS

### 2.1 DATA GENERATIVE PROCESS

For this scenario, we consider inputs $x \in \mathcal{X}$, targets $y \in \mathcal{Y}$, concepts $c = \{c_j\}_{j=1}^m \in \mathcal{C}$ and nuisances $n \in \mathcal{N}$, such that $p(x, y, c, n) = p(x|c, n)p(y|x)p(c, n)$, i.e., the inputs $x$ are described by the concepts $c$ and the nuisances $n$, and the targets $y$ are fully described by the input $x$. The only difference between $c$ and $n$ is that the former are observed, while the latter are not or, in other words, labels on $c$ are provided to us. We assume access to a training set $\{x^{(i)}, y^{(i)}, c^{(i)}\}_{i=1}^N$, which defines the empirical distribution $p(x, y, c) = \sum_{i=1}^N \delta\left(x - x^{(i)}\right) \delta\left(y - y^{(i)}\right) \delta\left(c - c^{(i)}\right)$. The graphical model corresponding to this generative process is shown in Figure 2a for the case of two concepts.

### 2.2 VANILLA MODELS

In machine learning, the most commonly studied problem is that of predicting $y$ from $x$, which serves as a foundation for more specialized tasks. We refer to models trained to address this problem as *Vanilla Models* (VMs). These models typically operate by first extracting an intermediate representation $z \in \mathcal{Z}$ from the input $x$ via an *encoder* $p_\theta(z \mid x)$ parameterized by $\theta$. Subsequently, a prediction $\hat{y} \in \mathcal{Y}$ is produced from $z$ using a *task head* $q_\phi(\hat{y} \mid z)$ parameterized by $\phi$. Since $z$ is intended to facilitate accurate prediction of $y$, the mutual information between $z$ and $y$, denoted $I(Z; Y)$, should be maximized. In Appendix B.1, we formally show that:

$$\max_Z I(Z; Y) = \max_{\theta, \phi} \mathbb{E}_{p(x, y)}\left[\mathbb{E}_{p_\theta(z|x)}\left[\log q_\phi(\hat{y}|z)\right]\right] \tag{1}$$

Figure 2b shows the graphical model of a Vanilla Model with a two-dimensional representation $\boldsymbol{z}$. Black edges represent the encoder $p_\theta(\boldsymbol{z} \mid \boldsymbol{x})$, while green edges indicate the task head $q_\phi(\hat{\boldsymbol{y}} \mid \boldsymbol{z})$. The *encoder* is typically chosen to be deterministic, i.e., $p_\theta(\boldsymbol{z} \mid \boldsymbol{x}) = \delta(\boldsymbol{z} - f_\theta(\boldsymbol{x}))$, where $f_\theta : \mathcal{X} \to \mathcal{Z}$ is a neural-network–parameterized mapping. However, for tractability reasons (see Section 2.4), we adopt a stochastic formulation where $p_\theta(\boldsymbol{z} \mid \boldsymbol{x}) = \mathcal{N}\left(\boldsymbol{z}; f_\theta(\boldsymbol{x}), \sigma_x^2 I\right)$, as summarized in Table 2. The choice of *task head* $q_\phi(\hat{\boldsymbol{y}} \mid \boldsymbol{z})$ depends on the structure of the output space $\mathcal{Y}$, also detailed in Table 2. The objective in Equation 1 corresponds to minimizing the cross-entropy loss when $\boldsymbol{y}$ is binary or multiclass, and to minimizing the mean squared error between $\boldsymbol{y}$ and $g_\phi^y(z)$ when $\boldsymbol{y}$ is continuous.

## 2.3 CONCEPT BOTTLENECK MODELS

Vanilla Models generally lack interpretability with respect to the known concepts $\boldsymbol{c}$, as the encoder $f_\theta$ is often opaque and difficult to analyze. Moreover, these models tend to entangle the concepts in such a way that it becomes intractable to determine how individual concepts influence specific components of the latent representation $\boldsymbol{z}$, and consequently the predictions $\hat{y}$. *Concept Bottleneck Models* (CBMs) have been introduced to address this limitation. In a CBM, each concept $c_j$ is predicted from a dedicated latent representation $z_j$ via a *concept head* $q(\hat{c}_j \mid z_j)$. Consequently, $z_j$ must encode all information about $c_j$—that is, $z_j$ must be a *sufficient* representation for $c_j$. This requirement can be formalized as maximizing the mutual information $I(Z_j; C_j)$, for which the following identity—proved in Appendix B.2—is employed:

$$\max_{Z_j} I(Z_j; C_j) = \max_{\theta, \phi} \mathbb{E}_{p(\boldsymbol{x}, c_j)} \left[ \mathbb{E}_{p_\theta(z_j \mid \boldsymbol{x})} \left[ \log q_\phi(\hat{c}_j \mid z_j) \right] \right] \tag{2}$$

As illustrated in Figure 2c, CBMs extend Vanilla Models by incorporating a concept head $q(\hat{c}_j \mid z_j)$, depicted with blue arrows. These models jointly optimize the objectives in Equations 1 and 2. As detailed in Table 2, the form of the concept head $q_\phi(\hat{c}_j \mid z_j)$ depends on the nature of the concept space $\mathcal{C}$. The objective in Equation 2 corresponds to minimizing the cross-entropy loss when $c_j$ is binary or multiclass, and the mean squared error between $c_j$ and $g_\phi^c(z_j)$ when $c_j$ is continuous.

**How are Interventions Performed in CBMs?** As explained in Section 1, a key advantage often attributed to CBMs is their ability to support concept-level interventions. Suppose we aim to estimate $p(\hat{\boldsymbol{y}} \mid c_j = \alpha, \boldsymbol{x})$. In CBMs, this intervention is performed through the latent representation $z_j$:

$$p(\hat{\boldsymbol{y}} \mid c_j = \alpha, \boldsymbol{x}) = \iint p(\hat{\boldsymbol{y}} \mid z_j, \boldsymbol{z}_{\backslash j}) p(z_j \mid c_j = \alpha) p(\boldsymbol{z}_{\backslash j} \mid \boldsymbol{x}) \, dz_j \, d\boldsymbol{z}_{\backslash j} \tag{3}$$

However, the conditional distribution $p(z_j \mid c_j)$ is not defined—there is no directed path from $c_j$ to $z_j$ in Figure 2c. Intuitively, because $z_j$ may encode information about $\boldsymbol{x}$ beyond $c_j$, it cannot be fully determined by $c_j$ alone. This raises a key question: *how can interventions be performed in CBMs if $p(z_j \mid c_j)$ is unknown?* To make interventions feasible, CBMs typically impose two constraints:

(i) Concepts $c_j \in \boldsymbol{c}$, are assumed to be binary. If a concept is originally multiclass with $k$ categories, it is converted into $k$ binary concepts (One-vs-Rest (Rifkin & Klautau, 2004)).

(ii) The *concept head* is defined as $q_\phi(c_j \mid z_j) = \sigma(z_j)$, where $\sigma$ denotes the sigmoid function.

Since $\sigma$ is invertible, this setup permits defining $p(z_j \mid c_j) \approx \sigma^{-1}(c_j)$. However, this is ill-defined at the binary extremes, as $\sigma^{-1}(1) = -\sigma^{-1}(0) = \infty$. To address this, in practice, $p(z_j \mid c_j = 0)$ and $p(z_j \mid c_j = 1)$ are set as the 5th and 95th percentiles of the empirical distribution of $z_j$, respectively (Koh et al., 2020). This workaround, however, introduces two crucial issues discussed in Section 5.

Table 2: Distributions considered in this work. $f_\theta : \mathcal{X} \to \mathcal{Z}$ is typically a large neural network while $g_\phi^y : \mathcal{Z} \to \mathcal{Y}$, $g_\phi^c : \mathcal{Z} \to \mathcal{C}$ and $g_\phi^z : \mathcal{C} \to \mathcal{Z}$ are comparatively lightweight networks (see Appendix F.1). Throughout this work, we model $\boldsymbol{z}$ as a continuous latent representation.

| | $p_\theta(z \mid x)$ | $q_\phi(\hat{y} \mid z)$ | $q_\phi(\hat{c}_j \mid z_j)$ | $q_\phi(\hat{z}_j \mid c_j)$ |
|---|---|---|---|---|
| Binary | - | Bernoulli $\left(g_\phi^y(z)\right)$ | Bernoulli $\left(g_\phi^c(z_j)\right)$ | - |
| Multiclass | - | Categoric $\left(g_\phi^y(z)\right)$ | Categoric $\left(g_\phi^c(z_j)\right)$ | - |
| Continuous | $\mathcal{N}\left(f_\theta(x), \sigma_x^2 I\right)$ | $\mathcal{N}\left(g_\phi^y(z), \sigma_{\hat{y}}^2 I\right)$ | $\mathcal{N}\left(g_\phi^c(z_j), \sigma_{\hat{c}}^2 I\right)$ | $\mathcal{N}\left(g_\phi^z(c_j), \sigma_{\hat{z}}^2 I\right)$ |

## 2.4 MINIMAL CONCEPT BOTTLENECK MODELS

As discussed in Section 1, CBMs lack an explicit mechanism for enforcing a bottleneck, which often undermines their intended advantages. To address this limitation, we introduce *Minimal Concept Bottleneck Models* (MCBMs), which explicitly impose an *Information Bottleneck*. This ensures that each $z_j$ retains all information about its associated concept $c_j$, while excluding information unrelated to $c_j$. In other words, $z_j$ becomes a *minimal sufficient* representation of $c_j$. To achieve this, we introduce a *representation head* $q_\phi(\hat{z}_j \mid c_j)$ that predicts $z_j$ from $c_j$. This encourages $z_j$ to discard information unrelated to $c_j$, as doing so improves the predictive accuracy of $q_\phi(\hat{z}_j \mid c_j)$. Formally, this objective corresponds to minimizing the conditional mutual information $I(Z_j; X \mid C_j)$. We note that if $z_j$ is a representation of $x$, then the following propositions are equivalent: (i) $I(Z_j; X|C_j) = 0$, (ii) the Markov Chain $X \leftrightarrow C_j \leftrightarrow Z_j$ is satisfied and (iii) $p(z_j|c_j) = p(z_j|x)$. To minimize $I(Z_j; X \mid C_j)$, we leverage the following identity, which is proven in Appendix B.3:

$$\min_{Z_j} I(Z_j; X|C_j) = \min_{\theta,\phi} \mathbb{E}_{p(x,c_j)} \left[ D_{KL} \left( p_\theta(z_j|x) || q_\phi(\hat{z}_j|c_j) \right) \right] \tag{4}$$

Figure 2d illustrates how MCBMs extend CBMs by introducing the representation head $q(\hat{z}_j \mid c_j)$, depicted with red arrows. These models are trained to jointly optimize the objectives given in Equations 1, 2, and 4. Although the KL Divergence in Equation 4 does not admit a closed-form solution in general, we show in Appendix B.4 that, under the distributional assumptions listed in Table 2, it reduces to the mean squared error between $f_\theta(x)$ and $g_\phi^z(c_j)$.

**How are Interventions Performed in MCBMs?** In contrast to CBMs, MCBMs explicitly constrain $z_j$ to contain only information about $c_j$. As a result, modifying $z_j$ corresponds to intervening solely on $c_j$. This is reflected in Figure 2d, where MCBMs introduce a directed path from $c_j$ to $z_j$ through the intermediate variable $\hat{z}_j$. This structure permits the computation of:

$$p(z_j|c_j) = \int p(z_j|\hat{z}_j) q_\phi(\hat{z}_j|c_j) \, d\hat{z}_j \tag{5}$$

which simplifies to $p(z_j \mid c_j) = q_\phi(z_j \mid c_j)$ when $p(z_j \mid \hat{z}_j) = \delta(z_j - \hat{z}_j)$, i.e., once the objective in Equation 4 is optimized and $z_j$ encodes exclusively $c_j$. As shown in Table 2, we define the encoder distribution as $q_\phi(z_j \mid c_j) = \mathcal{N}(g_\phi^z(c_j), \sigma_{\hat{z}}^2 I)$, where mean function $g_\phi^z(c_j)$ is chosen according to the following rules:

  (i) For binary concepts, $g_\phi^z(c_j) = \lambda$ if $c_j = 1$, and $g_\phi^z(c_j) = -\lambda$ otherwise.

  (ii) For categorical concepts, $g_\phi^z(c_j) = \lambda \cdot \text{one\_hot}(c_j)$. This mirrors the *Prototypical Learning* (Snell et al., 2017) approach where class-dependent prototypes $\{g_\phi^z(c_j)\}_{j=1}^m$ are fixed.

  (iii) For continuous concepts, $g_\phi^z(c_j) = \lambda \cdot c_j$.

Here, $\lambda$ is a scaling constant that controls the norm of the latent representation, fixed to $\lambda = 3$ in all experiments. Regarding the variance term, we set: (i) $\sigma_x = 0$ in the case of CBMs to obtain a deterministic encoder in line with their original formulation, and (ii) $\sigma_x = \sigma_{\hat{z}} = 1$ for MCBMs.

**Practical Considerations for Optimizing MCBMs** To optimize MCBMs, we combine the three previously introduced objectives, incorporating two key considerations. First, we approximate the expectations over $p(x, y)$ and $p(x, c_j)$ using the empirical data distribution, replacing integrals with summations over the dataset. Second, to enable gradient-based optimization through the stochastic encoder, we apply the reparameterization trick (Kingma, 2013): $\mathbb{E}_{p_\theta(z|x)} \left[ \log q_\phi(y|z) \right] \approx \sum_i \log q_\phi \left( y | f_\theta' \left( x, \epsilon^{(i)} \right) \right)$ and $E_{p_\theta(z_j|x)} \left[ \log q_\phi(\hat{c}_j|z_j) \right] \approx \sum_i \log q_\phi \left( \hat{c}_j | f_{\theta,j}' \left( x, \epsilon^{(i)} \right) \right)$, where $f_\theta'(x, \epsilon) = f_\theta(x) + \sigma_x^2 I \epsilon$ (due to the choice of $p_\theta(z|x)$ in Table 2), $\epsilon \sim \mathcal{N}(0, I)$ and $f_{\theta,j}'(x, \epsilon)$ corresponds to the element $j$ of $f_\theta'(x, \epsilon)$. Combining the considerations above yields the final objective for training MCBMs, as shown in Equation 6, where $\beta$ and $\gamma$ are hyperparameters. The first term corresponds to the objective used in Vanilla Models, the second term is introduced in CBMs, and the third is specific to MCBMs. Detailed training algorithms for the various cases listed in Table 2 are provided in Appendix C.

$$\max_{\theta,\phi} \sum_{k=1}^N \sum_i \log q_\phi \left( \hat{y} | f_\theta' \left( x^{(k)}, \epsilon^{(i)} \right) \right) + \beta \sum_{j=1}^n \log q_\phi \left( \hat{c}_j | f_{\theta,j}' \left( x^{(k)}, \epsilon^{(i)} \right) \right)$$
$$- \gamma \sum_{j=1}^n D_{KL} \left( p_\theta \left( z_j | x^{(k)} \right) || q_\phi \left( \hat{z}_j | c_j^{(k)} \right) \right) \tag{6}$$

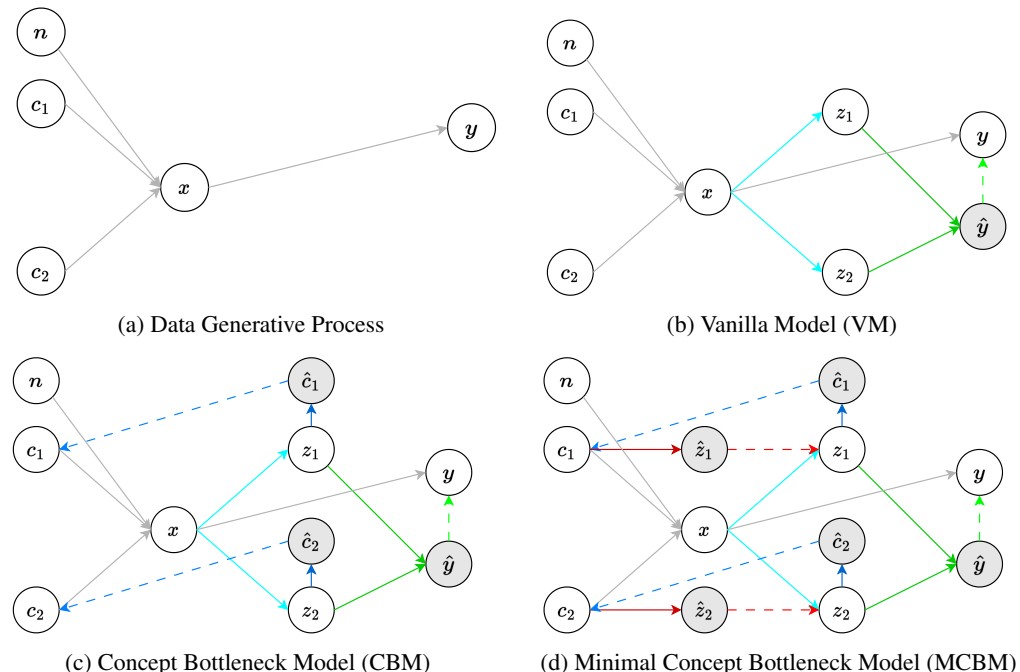

Figure 2: Graphical models of the different systems described for two concepts and two-dimensional representations. Appendix A shows the analogous figure for $m$ concepts and $m$-dimensional representations. Inputs $\boldsymbol{x}$ are defined by some concepts $\{c_j\}_{j=1}^m$ and nuisances $\boldsymbol{n}$; and targets $\boldsymbol{y}$ are defined by $\boldsymbol{x}$ (gray arrows). Vanilla models obtain the representations $\{z_j\}_{j=1}^m$ from $\boldsymbol{x}$ through the *encoder* $p_\theta(\boldsymbol{z}|\boldsymbol{x})$ (cyan arrows) and solve the task $\hat{\boldsymbol{y}}$ sequentially through the *task head* $q_\phi(\hat{\boldsymbol{y}}|\boldsymbol{z})$ (green arrows). Concept Bottleneck Models make a prediction $\hat{c}_j$ of each concept $c_j$ from one representation $z_j$ through the *concept head* $q_\phi(\hat{c}_j|z_j)$ (blue arrows). Minimal CBMs make a prediction $\hat{z}_j$ of each representation $z_j$ from one concept $c_j$ through the *representation head* $q_\phi(\hat{z}_j|c_j)$ (red arrows).

## 3 RELATED WORK

**Information Leakage in Concept Bottleneck Models** Information leakage occurs when the learned representation $\boldsymbol{z}$ encodes information outside the concept set $\boldsymbol{c}$, reducing both interpretability and intervenability (Margeloiu et al., 2021; Mahinpei et al., 2021). This arises when the Markovian assumption fails—i.e., when the target $\boldsymbol{y}$ is not fully determined by the concept set $\boldsymbol{c}$, or equivalently, $p(\boldsymbol{y}|\boldsymbol{c}) \neq p(\boldsymbol{y}|\boldsymbol{c}, \boldsymbol{x})$, which is typically the case in real-world scenarios (Havasi et al., 2022). Concept Embedding Models (CEMs) (Espinosa Zarlenga et al., 2022) were proposed to mitigate the accuracy–interpretability trade-off in CBMs. However, this trade-off is fundamentally limited by the chosen concept set, and CEMs may even be less interpretable than standard CBMs, as their more entropic representations tend to amplify information leakage, a critique formalized in (Parisini et al., 2025). Havasi et al. (2022) also introduced Hard Concept Bottleneck Models (HCBMs), which predict $\boldsymbol{y}$ from binarized concept predictions $\hat{c}_j$ rather than from $\boldsymbol{z}$ (see Appendix D), thereby imposing an ad-hoc Information Bottleneck. Extensions such as Autoregressive CBMs (ARCBMs) and Stochastic CBMs (SCBMs)(Havasi et al., 2022; Vandenhirtz et al., 2024) incorporate dependencies between concepts. Energy-based CBMs (Xu et al., 2024a) replace the task head with an energy function that scores concept–label compatibility, enabling richer and more structured concept relationships. Beyond architectural changes, prior work has characterized leakage using information-theoretic quantities (Parisini et al., 2025; Makonnen et al., 2025). To the best of our knowledge, this is the first work that leverages Information Theory to (i) formally identify the underlying design flaw—CBMs require each $z_j$ to predict $c_j$ but do not restrict $z_j$ from encoding additional nuisance information—and (ii) introduce a principled, variational IB objective that directly constrains each latent variable to retain only concept-relevant information.

**Information Bottleneck in Representation Learning** The Information Bottleneck (IB) (Tishby et al., 2000) provides a principled way to balance preserving information about a factor with compressing the representation. In this framework, a representation is *sufficient* if it retains all the information

about the variable of interest, and *minimal* if it contains only that information (Achille & Soatto, 2018a;b; Shwartz Ziv & LeCun, 2024). Computation of these quantities is intractable, so variational methods have been developed to derive tractable evidence bounds (Alemi et al., 2016; Fischer, 2020).

# 4 EXPERIMENTS

In this section, we present a series of experiments designed to empirically demonstrate that CBMs fail to impose an effective bottleneck, even in simple settings. We also examine the consequences of this limitation. In contrast, we show that MCBMs successfully enforce a bottleneck, thereby mitigating the issues that arise in its absence. Further implementation details—including encoder architectures and training hyperparameters for each experiment—are provided in Appendix F.

## 4.1 DO CBMS AND MCBMS LEAK INFORMATION?

Some works assume that the task $y$ is fully defined by the concepts $c$. However, it is unrealistic to expect a finite set of human-understandable concepts to completely describe an arbitrarily complex task. In practice, certain nuisances $n$ that describe the input $x$ also influence $y$. Specifically, we decompose the nuisances as $n = \{n_y, n_{\bar{y}}\}$, where $n_y$ captures nuisances that, together with the concepts, describe $y$, while $n_{\bar{y}}$ comprises those independent of $y$ (see Figure 3a). As discussed throughout this work, CBMs provide no incentive to remove information unrelated to the concepts. Moreover, since the objective of most models is to solve $y$, they are incentivized to preserve not only $c$ but also

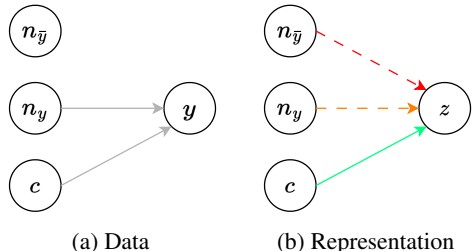

(a) Data      (b) Representation

Figure 3: Some nuisances $n_y \in n$ affect the task $y$ while others $n_{\bar{y}} \in n$ do not. None of them should affect the representation $z$ since it must be fully described by the concepts $c$.

$n_y$. While this may improve task performance, it comes at the expense of interpretability and valid interventions. For instance, if $z_j \in z$ is intended to represent the concept $c_j \in c$, , one might assume that modifying $z_j$ corresponds solely to intervening on $c_j$. However, if $z_j$ also encodes information about $n_y$ (or even $n_{\bar{y}}$ as we show later), then modifying $z_j$ also affects the nuisances, invalidating any causal conclusions. By introducing the Information Bottleneck in Equation 4, we explicitly constrain the model to remove nuisances from $z_j$, thereby restoring the validity of causal analyses involving $c_j$. We note, however, that this necessarily reduces task performance: if $c$ is incomplete and solving $y$ requires information from $n$ (i.e., $n_y \neq \emptyset$), excluding $n$ from $z$ will lower predictive performance.

We next examine whether $n_y$ and $n_{\bar{y}}$ are present in the representations across different CBM variants and datasets. For this purpose, we define the following task–concept configurations: (i) **MPI3D** (Gondal et al., 2019), where $y$ is the *object shape*, $n_y$ the *horizontal axis*, $n_{\bar{y}}$ the *vertical axis*, and $c$ the remaining generative factors; (ii) **Shapes3D** (Kim & Mnih, 2018), where $y$ is the *shape*, $n_y$ the *floor color* and *wall color*, $n_{\bar{y}}$ the *orientation*, and $c$ the remaining factors; (iii) **CIFAR-10** (Krizhevsky et al., 2009), where $y$ is the standard classification task, $c$ consists of 64 of the 143 attributes extracted by Oikarinen et al. (2023) using GPT-3 (Brown et al., 2020), and $n_y$ the remaining attributes, since all nuisances are correlated with $y$; (iv) **CUB** (Wah et al., 2011), where $y$ is the *bird species*, $c$ includes concepts from twelve randomly selected attribute groups, and $n_y$ the attributes from the remaining 15 groups; and (v) **AwA2** (Xian et al., 2017), where $y$ is the *animal class*, $c$ consists of 20 of the 85 human-annotated attributes, and $n_y$ the remaining attributes. While MCBMs natively support multiclass concepts, the other baselines in our study are limited to binary concepts. To ensure fairness, factors with $k$ classes are therefore represented as $k$ binary concepts.

**Task-related information leakage**    To measure the presence of a nuisance factor $n_j \in n_y$ in $z$, we estimate $I(N_j; Z \mid C)$—the information about $n_j$ contained in $z$ beyond what is explained by $c$. Since this quantity is intractable, we approximate it as $\hat{I}(N_j; Z|C) = \hat{H}(N_j|C) - \hat{H}(N_j|C, Z)$, where $\hat{H}(N_j \mid C) = -\sum_{k=1}^{N} \log h_\psi^c(c^{(k)})$ and $\hat{H}(N_j \mid C, Z) = -\sum_{k=1}^{N} \log h_\psi^{cz}(c^{(k)}, z^{(k)})$. Here, $h_\psi^c$ and $h_\psi^{cz}$ are MLP classifiers trained to predict $n_j$ from $c$ and $(c, z)$, respectively. In Table 3, we report the average value of $\frac{\hat{I}(N_j;Z|C)}{H(N_j)}$ across all $n_j \in n_y$, which we call *Uncertainty Reduction Ratio* (URR). From these results, we conclude that: (i) CBMs tend to reduce nuisance information compared to VMs, though not consistently; (ii) CEMs and ECBMs generally preserve the largest amount of nuisance information, often exceeding even VMs; (iii) ARCBMs and HCBMs show no systematic

advantage over CBMs in terms of nuisance removal; and (iv) MCBMs provide the strongest reduction of nuisance information, particularly as $\gamma$ increases, which enforces a stricter bottleneck.

Table 3: Average value of URR for task-related nuisances $\boldsymbol{n}_y$.

|  | MPI3D | Shapes3D | CIFAR-10 | CUB | AwA2 |
|---|---|---|---|---|---|
| Vanilla | $35.0 \pm 1.9$ | $45.5 \pm 6.4$ | $19.8 \pm 0.7$ | $3.8 \pm 1.0$ | $1.5 \pm 0.3$ |
| CBM | $28.1 \pm 0.5$ | $18.1 \pm 2.9$ | $18.5 \pm 0.7$ | $3.8 \pm 0.8$ | $1.4 \pm 0.3$ |
| CEM | $43.2 \pm 5.2$ | $15.8 \pm 3.9$ | $27.2 \pm 0.8$ | $3.9 \pm 1.1$ | $1.1 \pm 0.5$ |
| ECBM | $25.2 \pm 3.0$ | $47.1 \pm 3.7$ | $18.1 \pm 0.5$ | $4.5 \pm 1.0$ | $1.1 \pm 0.3$ |
| ARCBM | $28.2 \pm 1.7$ | $18.4 \pm 2.1$ | $18.2 \pm 0.6$ | $3.9 \pm 0.9$ | $1.6 \pm 0.3$ |
| SCBM | $24.3 \pm 0.4$ | $21.8 \pm 1.4$ | $18.3 \pm 0.7$ | $3.6 \pm 0.8$ | $1.2 \pm 0.2$ |
| MCBM (low $\gamma$) | $10.7 \pm 0.1$ | $2.5 \pm 0.1$ | $18.0 \pm 0.5$ | $3.4 \pm 0.9$ | $1.0 \pm 0.4$ |
| MCBM (medium $\gamma$) | $6.7 \pm 0.2$ | $0.2 \pm 0.3$ | $18.1 \pm 0.5$ | $2.8 \pm 0.8$ | $0.9 \pm 0.4$ |
| MCBM (high $\gamma$) | $\mathbf{0.0 \pm 0.0}$ | $\mathbf{0.0 \pm 0.0}$ | $\mathbf{17.6 \pm 0.5}$ | $\mathbf{2.4 \pm 1.0}$ | $\mathbf{0.7 \pm 0.4}$ |

**Task-unrelated information leakage** Neural networks typically discard input information, as their layers are non-invertible (Tishby & Zaslavsky, 2015; Tschannen et al., 2019). Moreover, since $\boldsymbol{n}_{\bar{y}}$ is irrelevant for predicting $\boldsymbol{y}$, there is no incentive to retain it in the representation $\boldsymbol{z}$. One might therefore expect $\boldsymbol{z}$ to be free of $\boldsymbol{n}_{\bar{y}}$. However, prior work has shown that neural representations often preserve information not directly related to the task (Achille & Soatto, 2018a; Arjovsky et al., 2019). To examine this, we analyze whether $\boldsymbol{n}_{\bar{y}}$ is present in $\boldsymbol{z}$ for MPI3D and Shapes3D—the only settings where $\boldsymbol{n}_{\bar{y}}$ is non-empty. Table 4 reports URR values for the

Table 4: Average value of URR for task- nuisances.

|  | MPI3D | Shapes3D |
|---|---|---|
| Vanilla | $11.3 \pm 0.1$ | $42.7 \pm 9.1$ |
| CBM | $7.4 \pm 1.9$ | $20.6 \pm 3.3$ |
| CEM | $15.5 \pm 4.2$ | $40.9 \pm 1.8$ |
| ECBM | $6.2 \pm 1.4$ | $46.4 \pm 4.7$ |
| ARCBM | $8.7 \pm 1.4$ | $26.6 \pm 0.5$ |
| SCBM | $7.0 \pm 0.3$ | $21.7 \pm 1.7$ |
| MCBM (l $\gamma$) | $\mathbf{0.0 \pm 0.0}$ | $\mathbf{0.0 \pm 0.0}$ |
| MCBM (m $\gamma$) | $\mathbf{0.0 \pm 0.0}$ | $\mathbf{0.0 \pm 0.0}$ |
| MCBM (h $\gamma$) | $\mathbf{0.0 \pm 0.0}$ | $\mathbf{0.0 \pm 0.0}$ |

nuisance variables in $\boldsymbol{n}_{\bar{y}}$ in. We find that: (i) as in Table 3, CEMs and ECBMs can retain more nuisance information than even VMs; and (ii) MCBMs are the only models that consistently eliminate nuisances across all values of $\gamma$, as expected: with no incentive to preserve $\boldsymbol{n}_{\bar{y}}$ and an explicit penalty for doing so, such information is naturally discarded.

**Are concepts removed to a greater extent in MCBMs?** One might worry that removing nuisance information could also inadvertently eliminate information about the concepts. To test this, we report concept prediction accuracy for CIFAR-10 and CUB in Table 5 (note that all models reach 100% accuracy on MPI3D and Shapes3D). We can observe that (i) no model consistently out-

Table 5: Average concepts accuracy

|  | CIFAR-10 | CUB | AwA2 |
|---|---|---|---|
| CBM | $84.8 \pm 0.2$ | $96.3 \pm 0.1$ | $98.1 \pm 0.1$ |
| CEM | $84.8 \pm 0.2$ | $96.3 \pm 0.1$ | $98.0 \pm 0.1$ |
| ECBM | $84.6 \pm 0.2$ | $96.1 \pm 0.4$ | $98.0 \pm 0.1$ |
| ARCBM | $84.3 \pm 0.2$ | $96.2 \pm 0.1$ | $97.9 \pm 0.1$ |
| SCBM | $84.3 \pm 0.2$ | $\mathbf{96.5 \pm 0.1}$ | $\mathbf{98.4 \pm 0.1}$ |
| MCBM (l $\gamma$) | $\mathbf{84.9 \pm 0.1}$ | $96.3 \pm 0.2$ | $98.0 \pm 0.2$ |
| MCBM (m $\gamma$) | $\mathbf{84.9 \pm 0.2}$ | $96.1 \pm 0.1$ | $97.9 \pm 0.1$ |
| MCBM (h $\gamma$) | $84.8 \pm 0.1$ | $95.8 \pm 0.3$ | $97.6 \pm 0.2$ |

performs the others—some preserve more concept information in CIFAR-10, while others perform slightly better in CUB; and (ii) increasing $\gamma$ in MCBMs gradually reduces concept accuracy, as stronger regularization may suppress features correlated with the concepts. These results indicate that MCBMs effectively remove nuisance information while largely preserving concept-relevant content.

**How does this affect task performance?** As previously discussed, when $\boldsymbol{n}_y$ is non-empty, restricting $\boldsymbol{z}$ to contain only information about $\boldsymbol{c}$ should reduce performance on $\boldsymbol{y}$: if $\boldsymbol{c}$ alone is insufficient to solve $\boldsymbol{y}$, then $\boldsymbol{z}$ cannot achieve perfect accuracy. In Table 6, we observe the following: (i) CBMs, CEMs and ECBMs reach task accuracy comparable to (or even higher than) VMs, indicating that they also rely on $\boldsymbol{n}_y$ to predict $\boldsymbol{y}$; (ii) ARCBMs and SCBMs achieve lower task accuracy, as they impose a bottleneck after the representations (see Appendix D); and (iii) MCBMs show decreasing task accuracy as $\gamma$ increases, reflecting the stricter bottleneck applied to the representations. Importantly, this reduction in task accuracy should not be viewed negatively: since all models achieve similar concept accuracy (see Table 5), it indicates that predictions rely less on nuisance information.

Table 6: Task accuracy

|  | MPI3D | Shapes3D | CIFAR-10 | CUB | AwA2 |
|---|---|---|---|---|---|
| Vanilla | $100.0 \pm 0.0$ | $100.0 \pm 0.0$ | $72.1 \pm 0.6$ | $77.4 \pm 0.3$ | $91.6 \pm 0.4$ |
| CBM | $99.9 \pm 0.0$ | $100.0 \pm 0.0$ | $73.8 \pm 0.1$ | $77.6 \pm 0.2$ | $91.1 \pm 0.4$ |
| CEM | $100.0 \pm 0.0$ | $100.0 \pm 0.0$ | $73.1 \pm 0.3$ | $77.5 \pm 0.3$ | $90.7 \pm 0.5$ |
| ECBM | $99.9 \pm 0.0$ | $100.0 \pm 0.0$ | $74.4 \pm 0.4$ | $77.1 \pm 0.5$ | $92.1 \pm 0.3$ |
| ARCBM | $24.2 \pm 0.6$ | $29.7 \pm 0.4$ | $68.9 \pm 0.3$ | $75.5 \pm 0.8$ | $89.9 \pm 0.3$ |
| SCBM | $24.2 \pm 0.5$ | $29.7 \pm 0.3$ | $62.3 \pm 0.0$ | $73.5 \pm 0.4$ | $91.8 \pm 0.1$ |
| MCBM (l $\gamma$) | $92.7 \pm 1.1$ | $100.0 \pm 0.0$ | $72.4 \pm 0.5$ | $78.3 \pm 0.6$ | $90.6 \pm 0.4$ |
| MCBM (m $\gamma$) | $46.0 \pm 0.6$ | $32.2 \pm 1.5$ | $70.8 \pm 0.2$ | $77.4 \pm 0.4$ | $90.1 \pm 0.2$ |
| MCBM (h $\gamma$) | $24.9 \pm 0.3$ | $30.1 \pm 0.2$ | $70.5 \pm 0.6$ | $73.5 \pm 1.5$ | $88.7 \pm 0.3$ |

## 4.2 Do MCBMs Yield More Interpretable Representations?

One of the key properties often attributed to CBMs is that their internal representations align with human-interpretable concepts (Debole et al., 2025). While this generally holds, we show that MCBMs yield representations that are even more interpretable. To support this claim, we employ two metrics: (i) *Centered Kernel Alignment* (CKA) (Cristianini et al., 2001; Cortes et al., 2012; Kornblith et al., 2019), which measures the similarity between learned representations and concept labels (encoded as one-hot vectors); (ii) *Disentanglement* (Eastwood & Williams, 2018), which assesses whether each dimension of the representation $z_j$ captures at most one concept $c_j$; and (iii) *Oracle Information Score* (OIS) (Zarlenga et al., 2023), which dextends beyond standard disentanglement by accounting for concept dependencies, requiring that correlations between $z_j$ and $z_k$ do not exceed those between $c_j$ and $c_k$. These metrics capture interpretability from complementary perspectives: (i) *CKA* indicates whether the information is organized in a concept-aware fashion; and (ii) Disentanglement and OIS reflect the extent to which individual concepts are independently encoded in the representations. As shown in Figure 4: As shown in Figure 4: (i) CBMs, CEMs, AR-CBMs, and HCBMs generally achieve higher CKA with concepts than Vanilla Models, but they do not consistently improve Disentanglement or OIS; (ii) ECBMs do not, in general, produce more interpretable representations than VMs; and (iii) MCBMs consistently improve CKA, Disentanglement, and OIS, with larger gains as $\gamma$ increases, reflecting the removal of additional nuisances.

These results indicate that explicitly removing nuisances leads to more interpretable representations. To analyze this systematically, Table 7 reports the rank correlations between the metrics in Figure 4 and the URR values in Table 3 across datasets. We observe that CKA and OIS are strong and consistent predictors of nuisance information, whereas disentanglement is noticeably weaker. This is particularly helpful for tuning $\gamma$: since $n_y$ is unobserved in real scenarios, URR cannot be computed as in Table 3. As a workaround, these metrics—computed solely from $c$ and $z$—can serve as practical proxies for selecting the trade-off between task accuracy and leakage.

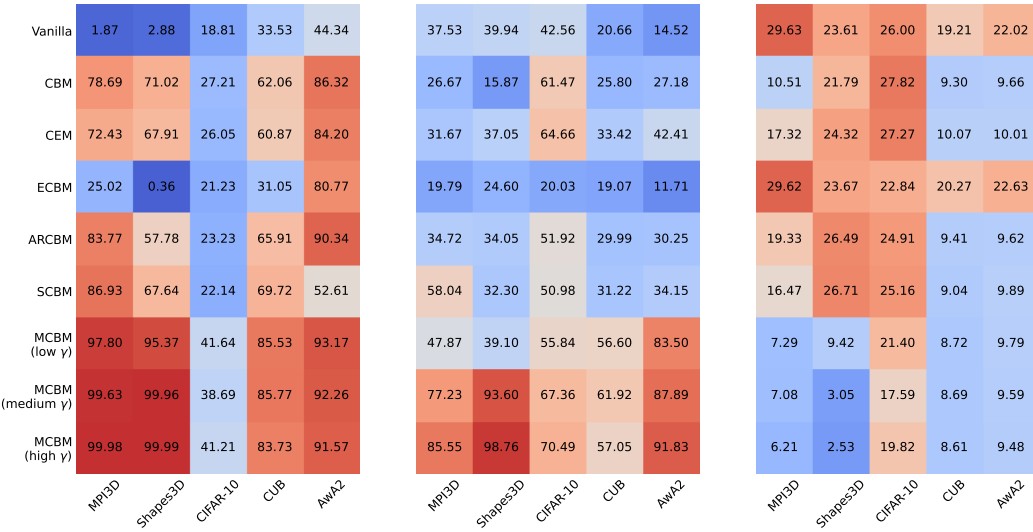

Figure 4: CKA (left, ↑), Disentanglement (middle, ↑) and OIS (right, ↓).

Table 7: Rank Correlation (p-value) between each metric and URR for $n_y$ across datasets

|  | MPI3D | Shapes3D | CIFAR-10 | CUB | AwA2 |
|---|---|---|---|---|---|
| CKA | $-.95\,(<.001)$ | $-.91\,(<.001)$ | $-.71\,(.04)$ | $-.77\,(.02)$ | $-.75\,(.04)$ |
| Disentanglement | $-.81\,(.01)$ | $-.57\,(.13)$ | $-.43\,(.29)$ | $-.69\,(.05)$ | $-.99\,(<.001)$ |
| OIS | $.90\,(.002)$ | $.81\,(.01)$ | $.86\,(.006)$ | $.89\,(.002)$ | $.66\,(.07)$ |

### 4.3 DO MCBMs ENABLE MORE RELIABLE INTERVENTIONS?

Another key property often attributed to CBMs is their capacity to support interventions. Although standard CBM intervention methods suffer from theoretical limitations (see Section 5), they can still be applied in practice. Accordingly, for CBMs we follow the procedure in Section 2.3, for ARCBMs and SCBMs we use the procedure described in Appendix D, and for MCBMs we apply the method in Section 2.4. To evaluate intervention reliability across models, we adopt the standardized protocol of Koh et al. (2020), which tracks how prediction error changes as the number of intervened concepts increases. Following Shin et al. (2023), we examine the effect of interventions using two complementary policies on CIFAR-10, CUB, and AwA2: (i) intervening on the concepts with the lowest predicted confidence (results in Figure 5); and (ii) intervening on randomly selected concepts (results in Appendix E). For MPI3D and Shapes3D, interventions yield only minor changes in performance, which aligns with Table 6 showing that their concepts are relatively weak predictors of the target. Across datasets where concepts are informative, both intervention policies lead to consistent and converging conclusions: (i) CBMs may even increase error when multiple concepts are intervened—an effect of nuisance information leaking into the representation due to the absence of a proper bottleneck; (ii) ARCBMs and SCBMs mitigate this by applying an Information Bottleneck after the representations, which improves intervention effectiveness while leaving interpretability unchanged (Figure 4); (iii) in MCBMs, intervention gains remain mostly insensitive to $\gamma$ when only a small or moderate number of concepts are intervened—evidenced by the nearly parallel curves across $\gamma$ values—but increase with higher $\gamma$ when more concepts are intervened, as reflected in the steeper negative slopes; (iv) at low and medium $\gamma$, MCBMs deliver the strongest intervention performance, especially when fewer than 100% of the concepts are intervened, clearly outperforming ARCBMs and SCBMs; and (v) for AwA2—where all models exhibit low levels of nuisance retention (Table 3)—intervention performance is nearly identical across methods.

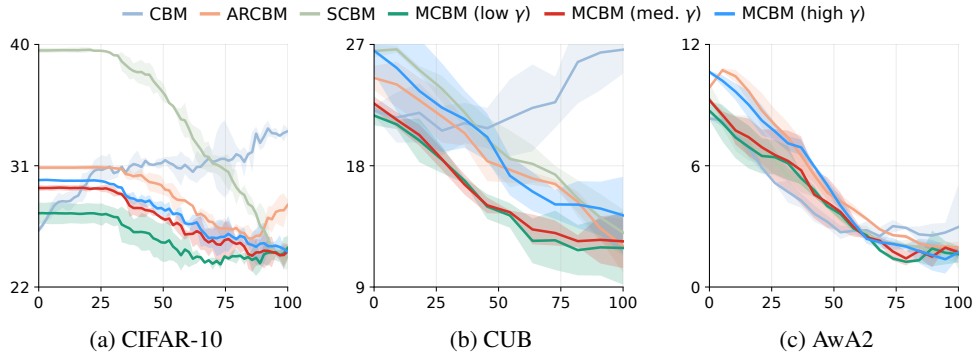

Figure 5: Error (y-axis) versus percentage of concepts intervened (x-axis) across different models.

### 5 OTHER FUNDAMENTAL THEORETICAL FLAWS OF CBMs

As briefly discussed in Section 2.3, CBMs—unlike MCBMs—do not provide a principled mechanism to estimate $p(z_j \mid c_j)$. To enable interventions, two assumptions are typically introduced: (i) multi-class concepts are handled using a One-vs-Rest scheme, and (ii) interventions are implemented via the sigmoid inverse function, i.e., $p(z_j \mid c_j) \approx \delta\left(z_j - \sigma^{-1}(c_j)\right)$. These assumptions are not only ad hoc but also theoretically incorrect, as we explain and illustrate with toy experiments below.

**One-vs-Rest Limitations** One-vs-Rest strategies exhibit several limitations: (i) individual binary classifiers tend to be biased toward the negative class, and (ii) the predicted probabilities across classifiers are typically uncalibrated (Bishop, 2006). To illustrate these issues, we design an experiment where the concepts are defined based on a four-class spiral dataset with imbalanced class distributions. We train: (i) a CBM with One-vs-Rest binarization, and (ii) an MCBM modeling concepts directly as multiclass variables.

The results reveal three major shortcomings of CBMs trained with the One-vs-Rest strategy. First, Figure 6 (left) shows that they overpredict the most frequent class, especially when the true class is rare and spatially close to the dominant. Second, Figure 6 (middle) shows that CBMs produce poorly calibrated predictions, lacking the smooth likelihood transitions observed in MCBMs. Finally, Figure 6 (right) shows that CBMs often assign near-one likelihoods to multiple classes simultaneously, whereas MCBMs confine high secondary likelihoods to regions of overlap, producing more reliable uncertainty estimates. Importantly, these effects may not be captured by standard metrics like accuracy, yet they represent fundamental flaws from a probabilistic perspective.

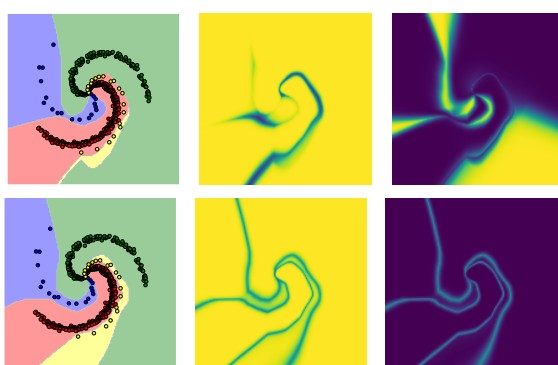

Figure 6: CBM (top) versus MCBM (bottom): class boundaries (left), highest predicted likelihood (middle), and second-highest predicted likelihood (right).

**Sigmoid Inverse Function** The intervention procedure $p(z_j \mid c_j) \approx \delta\left(z_j - \sigma^{-1}(c_j)\right)$ does not satisfy Bayes' rule, i.e., $p(z_j \mid c_j) = \frac{q_\phi(c_j \mid z_j) p(z_j)}{p(c_j)}$. For example, as illustrated in Figure 7, when the prior $p(z_j)$ is bimodal—a common case for representations arising from two classes—$p(z_j \mid c_j)$ differs markedly from $\sigma^{-1}(c_j)$. Ignoring the prior therefore not only violates probability theory but also yields poor practical approximations. This issue is difficult to overcome in CBMs, as the prior $p(z_j)$ is unknown and challenging to estimate.

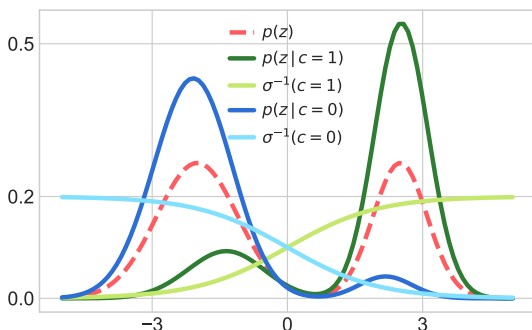

Figure 7: Density (y-axis) vs. $z$ (x-axis)

## 6  CONCLUSIONS, LIMITATIONS AND FUTURE DIRECTIONS

In this paper, we argue that—contrary to common belief—Concept Bottleneck Models (CBMs) do not enforce a true bottleneck: although representations are encouraged to retain concept-related information, they are not constrained to discard nuisance information. This limitation undermines interpretability and provides no theoretical guarantees for intervention procedures. To address this, we propose *Minimal Concept Bottleneck Models* (MCBMs), which introduce an Information Bottleneck in the representation space via an additional loss term derived through variational approximations. Beyond enforcing a proper bottleneck, our formulation ensures that interventions remain consistent with Bayesian principles. Empirically, we show that CBMs and their variants fail to remove non-concept information from the representation, even when irrelevant to the target task. In contrast, MCBMs effectively eliminate such information while preserving concept-relevant content, yielding more interpretable representations and principled interventions. Finally, we highlight fundamental flaws in the intervention process of CBMs, which MCBMs overcome due to their principled design.

Regarding the *limitations* of this work, we highlight the following: (i) MCBMs introduce a new hyperparameter, $\gamma$, which must be tuned to balance predictive accuracy and interpretability; (ii) the representation head $g_\phi^z$ adds a small number of parameters, though this overhead is negligible relative to the backbone; and (iii) while MCBMs consistently reduce nuisance information, complete removal remains challenging for high-variance datasets, suggesting that performance depends on the expressive capacity of the backbone architecture.

As for *future directions*, promising avenues include: (i) extending the model with an auxiliary latent variable $z_{m+1}$ appended to $\boldsymbol{z}$ to explicitly capture task-relevant nuisance factors $n_y$, allowing $z_1, \ldots, z_m$ to remain strictly interpretable while still retaining task-useful information in the full representation $\boldsymbol{z}$; and (ii) studying how the choice of prior distributions $q_\phi(z_j \mid c_j)$ affects representation quality and evaluation metrics such as disentanglement, alignment, and concept leakage.

ACKNOWLEDGMENTS

This work has received funding from the European Union's Horizon 2020 research and innovation programme under the Marie Skłodowska-Curie grant agreement No 101007666, MCIN/AEI/10.13039/501100011033 under Grant PID2024-155948OB-C53, and the Government of Aragón (Grant Group T36 23R).

CODE AVAILABILITY

An implementation accompanying this work is available at `https://github.com/antonioalmudevar/minimal_cbm`

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

## A   COMPLETE GRAPHICAL MODELS

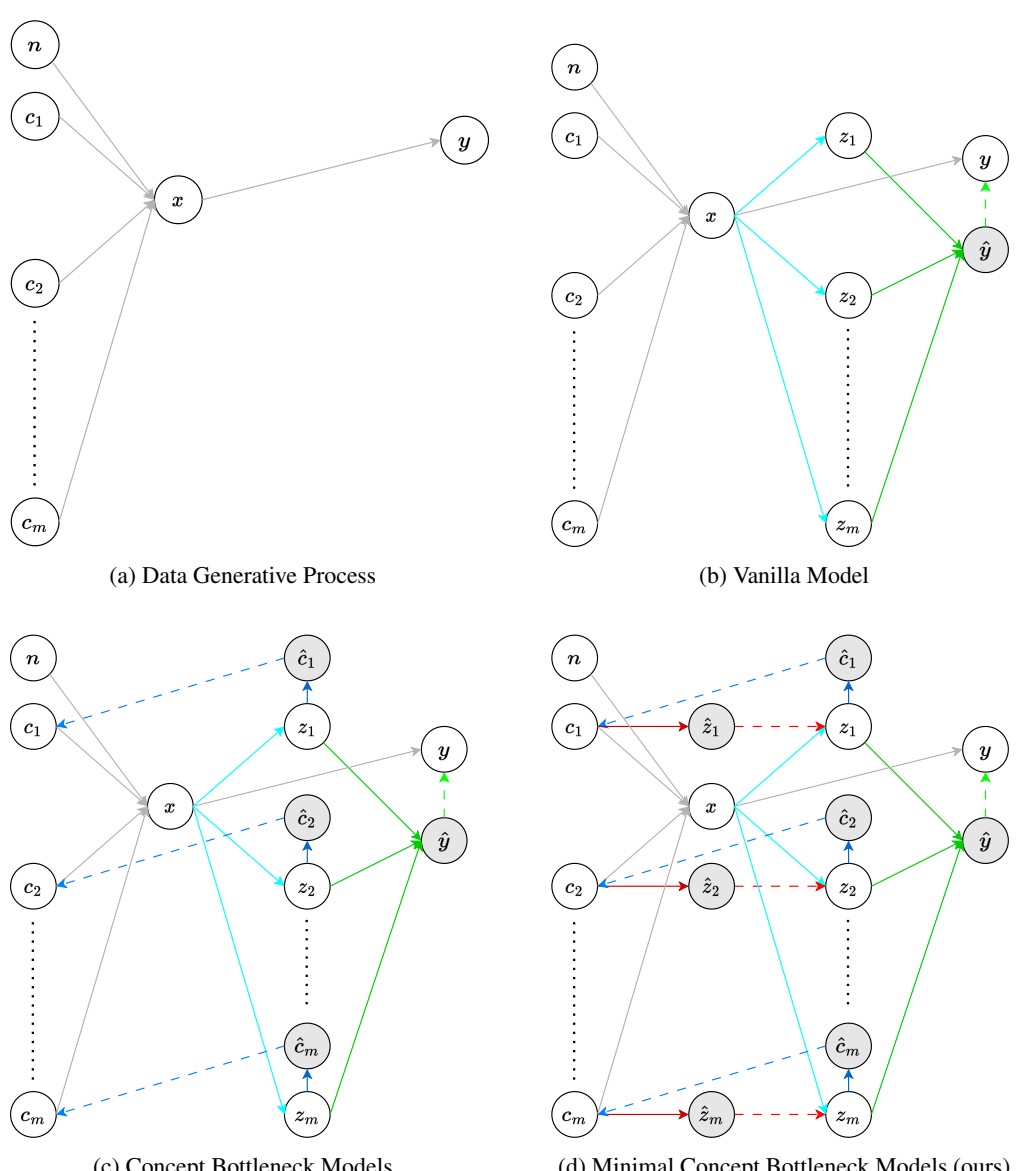

(a) Data Generative Process

(b) Vanilla Model

(c) Concept Bottleneck Models

(d) Minimal Concept Bottleneck Models (ours)

Figure 8: Graphical models of the different systems described. We consider $m$ concepts and $m$-dimensional representations. Inputs $x$ are defined by some concepts $\{c_j\}_{j=1}^m$ and nuisances $n$; and targets $y$ are defined by $x$ (gray arrows). Vanilla models obtain the representations $\{z_j\}_{j=1}^m$ from $x$ through the *encoder* $p_\theta(z|x)$ (cyan arrows) and solve the task $\hat{y}$ sequentially through the *task head* $q_\phi(\hat{y}|z)$ (green arrows). Concept Bottleneck Models make a prediction $\hat{c}_j$ of each concept $c_j$ from one representation $z_j$ through the *concept head* $q(\hat{c}_j|z_j)$ (blue arrows). Minimal CBMs make a prediction $z_j$ of each representation $z_j$ from one concept $c_j$ through the *representation head* $q(z_j|c_j)$ (red arrows).

# B   DETAILS OF DERIVATIONS

## B.1   PROOF OF EQUATION 1

$$I(Z;Y) = \iint p(z,y) \log \frac{p(y|z)}{p(y)} \, dy \, dz \tag{7}$$

$$= \iiint p(x,y) p_\theta(z|x) \log \frac{p(y|z)}{p(y)} \, dx \, dy \, dz \tag{8}$$

$$= \iiint p(x,y) p_\theta(z|x) \log \frac{p(y|z)}{p(y)} \frac{q_\phi(\hat{y}|z)}{q_\phi(\hat{y}|z)} \, dx \, dy \, dz \tag{9}$$

$$= \mathbb{E}_{p(x,y)} \left[ \mathbb{E}_{p_\theta(z|x)} \left[ \log q_\phi(\hat{y}|z) \right] \right] + E_{p_\theta(z)} \left[ D_{KL} \left( p(y|z) || q_\phi(\hat{y}|z) \right) \right] + H(Y) \tag{10}$$

$$\geq \mathbb{E}_{p(x,y)} \left[ \mathbb{E}_{p_\theta(z|x)} \left[ \log q_\phi(\hat{y}|z) \right] \right] + H(Y) \tag{11}$$

Thus, since $H(Y)$ is independent of $\theta$ and $\phi$, we have that:

$$\max_Z I(Z;Y) = \max_{\theta,\phi} \mathbb{E}_{p(x,y)} \left[ \mathbb{E}_{p_\theta(z|x)} \left[ \log q_\phi(\hat{y}|z) \right] \right] \tag{12}$$

## B.2   PROOF OF EQUATION 2

$$I(Z_j;C_j) = \iint p(z_j,c_j) \log \frac{p(c_j|z_j)}{p(c_j)} \, dc_j \, dz_j \tag{13}$$

$$= \iiint p(x,c_j) p_\theta(z_j|x) \log \frac{p(c_j|z_j)}{p(c_j)} \, dx \, dc_j \, dz_j \tag{14}$$

$$= \iiint p(x,c_j) p_\theta(z_j|x) \log \frac{p(c_j|z_j)}{p(c_j)} \frac{q(\hat{c}_j|z_j)}{q(\hat{c}_j|z_j)} \, dx \, dc_j \, dz_j \tag{15}$$

$$\geq E_{p(x,c_j)} \left[ E_{p_\theta(z_j|x)} \left[ \log q(\hat{c}_j|z_j) \right] \right] + E_{p_\theta(z_j)} \left[ D_{KL} \left( p(c_j|z_j) || q(\hat{c}_j|z_j) \right) \right] + H(C_j) \tag{16}$$

$$\geq E_{p(x,c_j)} \left[ E_{p_\theta(z_j|x)} \left[ \log q(\hat{c}_j|z_j) \right] \right] + H(C_j) \tag{17}$$

Given the fact that $H(C_j)$ is constant, we have that:

$$\max_{Z_j} I(Z_j;C_j) = \max_\theta E_{p(x,c_j)} \left[ E_{p_\theta(z_j|x)} \left[ \log q(\hat{c}_j|z_j) \right] \right] \tag{18}$$

## B.3   PROOF OF EQUATION 4

$$I(Z_j;X|C_j) = \iiint p(x,c_j) p_\theta(z_j|x) \log \frac{p(z_j|x)}{p(z_j|c_j)} \, dx \, dc_j \, dz_j \tag{19}$$

$$= \iiint p(x,c_j) p_\theta(z_j|x) \log \frac{p(z_j|x)}{p(z_j|c_j)} \frac{q(\hat{z}_j|c_j)}{q(\hat{z}_j|c_j)} \, dx \, dc_j \, dz_j \tag{20}$$

$$= \mathbb{E}_{p(x,c_j)} \left[ D_{KL} \left( p_\theta(z_j|x) || q(\hat{z}_j|c_j) \right) \right] - E_{p(c_j)} \left[ D_{KL} \left( p(z_j|c_j) || q(\hat{z}_j|c_j) \right) \right] \tag{21}$$

$$\leq \mathbb{E}_{p(x,c_j)} \left[ D_{KL} \left( p_\theta(z_j|x) || q(\hat{z}_j|c_j) \right) \right] \tag{22}$$

Thus, we have that:

$$\min_{Z_j} I(Z_j;X|C_j) = \min_\theta E_{p(x,c_j)} \left[ D_{KL} \left( p_\theta(z_j|x) || q(\hat{z}_j|c_j) \right) \right] \tag{23}$$

### B.4 KL DIVERGENCE BETWEEN TWO GAUSSIAN DISTRIBUTIONS

We are given the conditional distributions:

$$p_\theta(z_j|x) = \mathcal{N}(f_\theta(x)_j, \sigma_x^2 I),$$
$$q_\phi(\hat{z}_j|c_j) = \mathcal{N}(g_\phi^z(c_j), \sigma_{\hat{z}}^2 I),$$

and aim to minimize the expected KL divergence:

$$\min_{\theta,\phi} \mathbb{E}_{p(x,c_j)} \left[ D_{\text{KL}} \left( p_\theta(z_j|x) \,\|\, q_\phi(\hat{z}_j|c_j) \right) \right].$$

The KL divergence between two multivariate Gaussians with diagonal covariances is given by:

$$D_{\text{KL}}(\mathcal{N}(\mu_p, \Sigma_p) \,\|\, \mathcal{N}(\mu_q, \Sigma_q)) = \frac{1}{2} \left[ \text{tr}(\Sigma_q^{-1} \Sigma_p) + (\mu_q - \mu_p)^\top \Sigma_q^{-1} (\mu_q - \mu_p) - d + \log \frac{\det \Sigma_q}{\det \Sigma_p} \right].$$

Applying this to our case:

- $\mu_p = f_\theta(x)_j$, $\mu_q = g_\phi^z(c_j)$,
- $\Sigma_p = \sigma_x^2 I$, $\Sigma_q = \sigma_{\hat{z}}^2 I$,
- $d$ is the dimension of $z_j$.

Plugging in, we obtain:

$$D_{\text{KL}} = \frac{1}{2} \left[ \frac{d\sigma_x^2}{\sigma_{\hat{z}}^2} + \frac{1}{\sigma_{\hat{z}}^2} \|f_\theta(x)_j - g_\phi^z(c_j)\|^2 - d + d \log \left( \frac{\sigma_{\hat{z}}^2}{\sigma_x^2} \right) \right].$$

Note that all terms except the squared distance are constant with respect to $\theta$ and $\phi$. Therefore:

$$\mathbb{E}_{p(x,c_j)} \left[ D_{\text{KL}} \left( p_\theta(z_j|x) \,\|\, q_\phi(\hat{z}_j|c_j) \right) \right] = \frac{1}{2\sigma_{\hat{z}}^2} \mathbb{E}_{p(x,c_j)} \left[ \|f_\theta(x)_j - g_\phi^z(c_j)\|^2 \right] + \text{const.}$$

**Conclusion:** Minimizing the expected KL divergence

$$\min_{\theta,\phi} \mathbb{E}_{p(x,c_j)} \left[ D_{\text{KL}} \left( p_\theta(z_j|x) \,\|\, q_\phi(\hat{z}_j|c_j) \right) \right]$$

is equivalent (up to a scaling factor) to minimizing the expected mean squared error:

$$\min_{\theta,\phi} \mathbb{E}_{p(x,c_j)} \left[ \|f_\theta(x)_j - g_\phi^z(c_j)\|^2 \right].$$

## C  TRAINING ALGORITHM OF MCBMS

---

**Algorithm 1** Training Algorithm for MCBMs

---

**Input:** Dataset $\mathcal{D} = \{\boldsymbol{x}^{(k)}, \boldsymbol{y}^{(k)}, \boldsymbol{c}^{(k)}\}_{k=1}^{N}$, latent norm $\lambda$, learning rate $\eta$, batch size $B$
**Output:** Parameters $\theta$ (encoder), $\phi$ (class-head, task-heads, representation-heads)

1: Initialize parameters $\theta$, $\phi$ and representations heads:
2: **for all** $j = 1, \ldots, n$ **do**
3:     **if** $c_j$ is binary **then**
4:         $g_j^z \leftarrow \lambda$ if $c_j = 1$ else $-\lambda$
5:     **else if** $c_j$ is multiclass **then**
6:         $g_j^z \leftarrow \lambda \cdot \text{one\_hot}(c_j)$
7:     **else**
8:         $g_j^z \leftarrow \lambda \cdot c_j$
9:     **end if**
10: **end for**
11: **while** not converged **do**
12:     Sample a mini-batch $\{\boldsymbol{x}^{(k)}, \boldsymbol{y}^{(k)}, \boldsymbol{c}^{(k)}\}_{k=1}^{B} \sim \mathcal{D}$
13:     **for all** $\boldsymbol{x}^{(k)}, \boldsymbol{y}^{(k)}, \boldsymbol{c}^{(k)}$ in batch **do**
14:         Encode: Compute $\mu_\theta^{(k)} \leftarrow f_\theta\left(x^{(k)}\right)$
15:         Sample noise $\epsilon \sim \mathcal{N}(0, I)$         ▷ Reparameterization trick with only one sample
16:         Reparameterize: $\boldsymbol{z}^{(k)} \leftarrow \mu_\theta^{(k)} + \sigma_x \odot \epsilon$
17:         Task prediction: $\hat{\boldsymbol{y}}^{(k)} \leftarrow g_\phi^y\left(\boldsymbol{z}^{(k)}\right)$         ▷ Similar to VMs
18:         Task loss: $\mathcal{L}_y^{(k)} \leftarrow \left\|\boldsymbol{y}^{(k)} - \hat{\boldsymbol{y}}^{(k)}\right\|^2$ if $\boldsymbol{y}$ is continuous else $\text{CE}\left(\boldsymbol{y}^{(k)}, \hat{\boldsymbol{y}}^{(k)}\right)$
19:         **for all** $j = 1, \ldots, n$ **do**
20:             Concept $j$ prediction: $\hat{c}_j^{(k)} \leftarrow g_{\phi,j}^c\left(z_j^{(k)}\right)$         ▷ Similar to CBMs
21:             Concept $j$ loss: $\mathcal{L}_{c,j}^{(k)} \leftarrow \left\|c_j^{(k)} - \hat{c}_j^{(k)}\right\|^2$ if $c_j$ is continuous else $\text{CE}\left(c_j^{(k)}, \hat{c}_j^{(k)}\right)$
22:             Representation $j$ prediction: $\hat{z}_j^{(k)} \leftarrow g_j^z\left(c_j^{(k)}\right)$         ▷ Novelty in MCBMs
23:             Representation $j$ loss: $\mathcal{L}_{z,j}^{(k)} \leftarrow \left\|z_j^{(k)} - \hat{z}_j^{(k)}\right\|^2$
24:         **end for**
25:         Total loss: $\mathcal{L}^{(k)} \leftarrow \mathcal{L}_y^{(k)} + \beta \sum_{j=1}^{n} \mathcal{L}_{c,j}^{(k)} + \gamma \sum_{j=1}^{n} \mathcal{L}_{z,j}^{(k)}$
26:     **end for**
27:     Update $\theta$, $\phi$ using gradient descent:

$$\theta \leftarrow \theta - \eta \nabla_\theta\left(\frac{1}{B} \sum_{k=1}^{B} \mathcal{L}^{(k)}\right), \quad \phi \leftarrow \phi - \eta \nabla_\phi\left(\frac{1}{B} \sum_{k=1}^{B} \mathcal{L}^{(k)}\right)$$

28: **end while**

---

# D   HARD CONCEPT BOTTLENECK MODELS

As discussed in Section 3, one of the most successful approaches to mitigating information leakage is the family of *Hard Concept Bottleneck Models* (HCBMs) (Havasi et al., 2022). Unlike MCBMs, which enforce an Information Bottleneck (IB) directly at the representation level $z$, HCBMs operate by pruning the information contained in $z$ before producing the task prediction $\hat{y}$. In doing so, they encourage $\hat{y}$ to depend solely on $c$, thereby yielding interventions that are more reliable in practice. This approach differs fundamentally from MCBMs: whereas Equation 4 in MCBMs minimizes over $Z_j$, thereby removing nuisances directly from the representation $z_j$, HCBMs optimize at the prediction level, focusing on $\hat{Y}$. Formally, they are trained to minimize:

$$\min_{\hat{Y}} I(\hat{Y}; X|C) = \min_{\theta,\phi} \mathbb{E}_{p(x,c_j)} \left[ D_{KL} \left( p_{\theta,\phi}(\hat{\boldsymbol{y}}|x) || p(\hat{\boldsymbol{y}}|c) \right) \right] \tag{24}$$

That is, the optimization is performed directly over the predictions $\hat{y}$. To implement this, it defines:

$$p_{\theta,\phi}(\hat{\boldsymbol{y}}|x) = \iiint q_\phi \left( \hat{\boldsymbol{y}} | \hat{\boldsymbol{c}}^b \right) p \left( \hat{\boldsymbol{c}}^b | \hat{\boldsymbol{c}} \right) q \left( \hat{\boldsymbol{c}} | \boldsymbol{z} \right) p_\theta \left( \boldsymbol{z} | \boldsymbol{x} \right) d\hat{\boldsymbol{c}}^b d\hat{\boldsymbol{c}} d\boldsymbol{z} \tag{25}$$

where $q_\phi \left( \hat{\boldsymbol{y}} \mid \hat{\boldsymbol{c}}^b \right)$ denotes the new *task head*, $q \left( \hat{\boldsymbol{c}} \mid \boldsymbol{z} \right)$ the *concept head*, and $p_\theta \left( \boldsymbol{z} \mid \boldsymbol{x} \right)$ the *encoder*, as defined in Section 2. The distribution

$$p \left( \hat{c}_j^b | \hat{c}_j \right) = \delta \left( \hat{c}_j^b - \Theta \left( \hat{c}_j - 0.5 \right) \right) \tag{26}$$

referred to as the *binarizing head*, applies the Heaviside step function $\Theta$ to produce a binary version $\hat{c}_j^b \in \{0, 1\}$ of $\hat{c}_j$. In this way, HCBMs enforce an ad-hoc Information Bottleneck by predicting $\hat{y}$ from binarized concept representations. This process is schematized in Figure 9.

**How are Interventions Performed in HCBMs?**   Because HCBMs introduce a bottleneck immediately before predicting $\hat{y}$, the intervention process is more straightforward than in standard CBMs. More specifically, interventions are carried out according to:

$$p(\hat{\boldsymbol{y}}|c_j = \alpha, \boldsymbol{x}) = \iint q_\phi(\hat{\boldsymbol{y}}|\hat{c}_j^b, \hat{\boldsymbol{c}}_{\setminus j}^b) p(\hat{c}_j^b|c_j = \alpha) p_\theta(\hat{\boldsymbol{c}}_{\setminus j}^b|\boldsymbol{x}) \, d\hat{c}_j^b \, d\hat{\boldsymbol{c}}_{\setminus j}^b \tag{27}$$

Here, $p(\hat{c}_j^b|c_j = \alpha)$ is approximated as $\delta(\hat{c}_j^b - \alpha)$, while the other distributions are available in closed form. Although this procedure provides stronger guarantees than the intervention mechanisms in standard CBMs, two main issues remain:

(i) The optimization is performed over the predictions $\hat{y}$ instead of the representation $z$. As a result, the representations themselves are not necessarily interpretable, as evidenced in Figure 4, which limits one of the core motivations for adopting CBMs in the first place.

(ii) HCBMs still require binarizing multiclass concepts, which introduces theoretical limitations and practical drawbacks, as further discussed in Section 5.

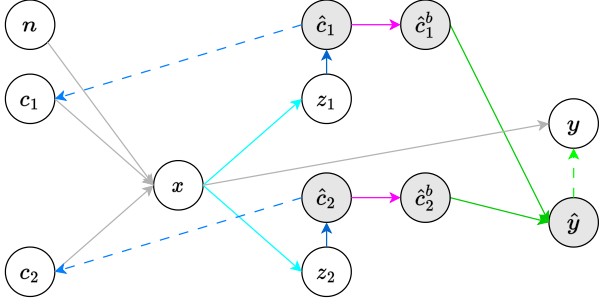

Figure 9: Graphical models of HCBMs with two concepts and two-dimensional representations. Hard Concept Bottleneck Models obtain a binarized version $\hat{c}_j^b$ of each predicted concept $\hat{c}_j$ through the *binarizing head* $p(\hat{c}_j^b \mid \hat{c}_j)$ (fuchsia arrows). Unlike the models in Figure 2, HCBMs predict the task output $\hat{y}$ from the binarized concepts $\hat{c}_j^b$ using the new *task head* $q_\phi(\hat{y} \mid \hat{c}^b)$ (green arrows).

## E    OTHER INTERVENTION PROCEDURES

As detailed in Section 4.3, we further examine how performance evolves as we intervene on an increasing number of randomly selected concepts, with the resulting curves shown in Figure 10. The overall behavior is consistent with the trends observed in Figure 5, highlighting similar model characteristics. As expected, the results exhibit somewhat higher variance across seeds, since each run intervenes on a different subset of concepts.

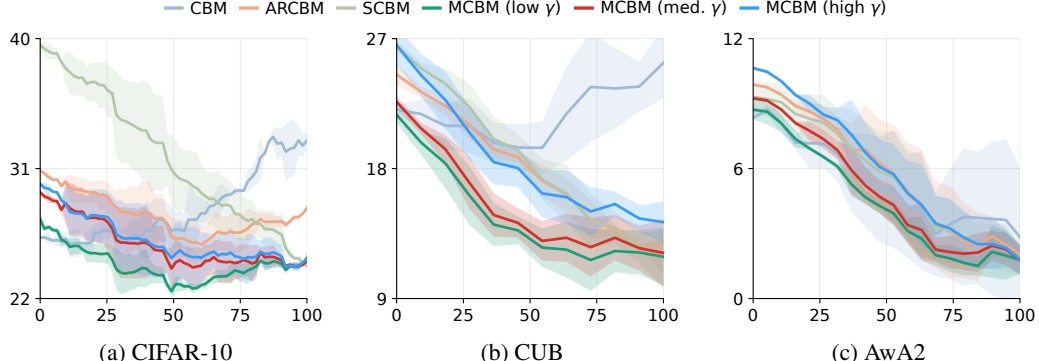

Figure 10: Error (y-axis) versus percentage of concepts intervened (x-axis) across different models for randomly selected concepts.

## F    EXPERIMENTS DETAILS

### F.1    HYPERPARAMETERS FOR SECTION 4

Table 8: Hyperparameters for Section 4. For all datasets, the concept head $g_\phi^c$ is implemented as the identity function in CBMs, and as a multilayer perceptron (MLP) with three hidden layers in MCBMs. We emphasize that CBMs are, by design, restricted to use invertible $g_\phi^c$ to enable intervention procedures.

| | MPI3D | Shapes3D | CIFAR-10 | CUB | AwA2 |
|---|---|---|---|---|---|
| $f_\theta$ architecture | ResNet20 | ResNet20 | 2 conv. layers | InceptionV3 | ResNet-50 |
| $f_\theta$ pretraining | None | None | None | ImageNet | ImageNet |
| $g_\phi^y$ hidden layers | 64 | 64 | 64 | 256 | 256 |
| $g_\phi^c$ hidden layers | None | None | None | None | None |
| $g_\phi^z$ hidden layers[1] | 3 | 3 | 3 | 3 | 3 |
| low $\gamma$ | 1 | 1 | 0.1 | 0.05 | 0.05 |
| medium $\gamma$ | 3 | 3 | 0.3 | 0.1 | 0.1 |
| high $\gamma$ | 5 | 5 | 0.5 | 0.3 | 0.3 |
| number of epochs | 50 | 50 | 200 | 250 | 120 |
| batch size | 128 | 128 | 128 | 128 | 128 |
| optimizer | SGD | SGD | Adam | SGD | SGD |
| learning rate | $6 \times 10^{-3}$ | $6 \times 10^{-3}$ | $1 \times 10^{-4}$ | $2 \times 10^{-2}$ | $2 \times 10^{-2}$ |
| momentum | 0.9 | 0.9 | 0. | 0.9 | 0.9 |
| weight decay | $4 \times 10^{-5}$ | $4 \times 10^{-5}$ | $4 \times 10^{-5}$ | $4 \times 10^{-5}$ | $4 \times 10^{-5}$ |
| scheduler | Step | Step | Step | Step | Step |
| step size (epochs) | 20 | 20 | 80 | 100 | 50 |
| scheduler $\gamma$ | 0.1 | 0.1 | 0.1 | 0.1 | 0.1 |

---

[1]Beyond the choice of $\gamma$, MCBMs introduce an additional design decision: the architecture of $g_\phi^z(c_j)$. In all our experiments, we implement this module as a small MLP with a single hidden layer of size 3, adding only 8 parameters per concept. Since concept sets typically contain at most 200 concepts, this corresponds to roughly 1600 additional parameters—negligible compared to the size of standard neural encoders $f_\theta$.

## F.2 DATASETS

**MPI3D** This is a synthetic dataset with controlled variation across seven generative factors: *object shape*, *object color*, *object size*, *camera height*, *background color*, *horizontal axis*, and *vertical axis*. In our setup, $y$ corresponds to the *object shape*, $n_y$ to the *horizontal axis*, $n_{\bar{y}}$ to the *vertical axis*, and $c$ to the remaining generative factors. To ensure consistency in the mapping between concepts and task nuisances and the target, we filter the dataset such that any combination of elements in $\{c, n_y\}$ corresponds to a unique value of $y$. All invalid combinations are removed accordingly.

**Shapes3D** This synthetic dataset consists of 3D-rendered objects placed in a room, with variation across six known generative factors: *floor color*, *wall color*, *object color*, *scale*, *shape*, and *orientation*. In our setup, $y$ corresponds to the *shape*, $ny$ includes *floor color* and *wall color*, $n_{\bar{y}}$ corresponds to *orientation*, and $c$ comprises the remaining factors. We follow the same filtering strategy as in MPI3D to construct this configuration: we retain only those samples for which each combination of $\{c, n_y\}$ uniquely determines $y$, removing all invalid configurations.

**CIFAR-10** CIFAR-10 is a widely used image classification benchmark consisting of 60,000 natural images of size $32 \times 32$, divided into 10 classes (e.g., airplanes, automobiles, birds, cats, etc.). The dataset is split into 50,000 training and 10,000 test images, with balanced class distributions. To reduce the need for manual concept annotations, the concepts are synthetically derived following the methodology of (Vandenhirtz et al., 2024). A total of 143 attributes are extracted using GPT-3 (Brown et al., 2020); 64 form the concept set $c$, while the rest define the nuisance set $n_y$. Binary values are obtained with the CLIP model (Radford et al., 2021) by comparing the similarity of each image to the embedding of an attribute and to its negative counterpart.

**CUB** The Caltech-UCSD Birds (CUB) dataset contains 11,788 images of 200 bird species, annotated with part locations, bounding boxes, and 312 binary attributes. Following the approach of Koh et al. (2020), we retain only the attributes that are present in at least 10 species (based on majority voting), resulting in a filtered set of 112 attributes. These attributes are grouped into 27 semantic clusters, where each group is defined by a common prefix in the attribute names. In our setup, the task variable $y$ is to classify the bird species. The concept set $c$ consists of the attributes belonging to 12 randomly selected groups (per run), while the nuisance set $n_y$ includes the attributes from the remaining 15 groups. Since most attributes exhibit some correlation with the classification task, we set $n_{\bar{y}}$ to the empty set.

**AwA2** The Animals with Attributes 2 (AwA2) dataset (Xian et al., 2017) contains 37,322 images of 50 animal classes annotated with 85 human-defined attributes describing appearance, behavior, and habitat. In our setup, the task variable $y$ is the *animal class*. To construct the concept and nuisance partitions, we retain a subset of 20 attributes—covering fundamental appearance and morphology features—as the concept set $c$, while the remaining 65 attributes form the nuisance set $n_y$. Since nearly all attributes exhibit some degree of correlation with the class label, we set $n_{\bar{y}}$ to the empty set. Following the preprocessing in Xian et al. (2017), we binarize continuous attributes using a threshold of 0.5.

