# OpenReview forum: "There Was Never a Bottleneck in Concept Bottleneck Models"
_ICLR.cc/2026/Conference — ICLR 2026 Poster_

### Official Review · Reviewer_6hx5 · 2025-10-22

**Soundness:** 3
**Presentation:** 3
**Contribution:** 2
**Rating:** 4
**Confidence:** 3

**Summary:**

This paper addresses the limitations of standard Concept Bottleneck Models (CBMs) in ensuring true concept-level interpretability. The authors argue that while CBMs predict human-defined concepts, their internal representations may still encode extraneous information, undermining interpretability and intervention validity. To address this, they propose Minimal Concept Bottleneck Models (MCBMs), which introduce an Information Bottleneck (IB) objective to ensure each representation component retains only concept-relevant information.

**Strengths:**

- The paper is well-organized, and the arguments regarding CBM limitations are presented clearly, with intuitive figures that effectively contrast CBMs and MCBMs.
- The paper provides formal justification for the proposed Information Bottleneck-based extension, connecting interpretability, information leakage, and probabilistic soundness.
- Experiments across multiple datasets and model variants (e.g., CBM, CEM, AR-CBM, SCBM) offer a broad empirical perspective, strengthening the claim that MCBMs improve both alignment and disentanglement.

**Weaknesses:**

- While the paper’s theoretical insights are valuable, the overall novelty appears modest. The Information Bottleneck principle is a well-established framework, and similar approaches applying it to CBMs seem to have been explored in [1] and [2]. Given this overlap, it would be helpful if the authors could better clarify the significance of their contribution—specifically, what distinguishes their approach from prior works.
- In Table 5, the accuracy reported for SCBM in [3] is 86.00, which appears higher than the 84.8/84.9 achieved by MCBM. Could the authors clarify what accounts for this difference?
- Introducing additional layers based on the Information Bottleneck framework would likely increase inference cost and memory usage. Could the authors quantify how much overhead this introduces?

**References** \
[1] Parisini, E., Chakraborti, T., Harbron, C., MacArthur, B. D., & Banerji, C. R. (2025). Leakage and interpretability in concept-based models. arXiv preprint arXiv:2504.14094. \
[2] Makonnen, M., Vandenhirtz, M., Laguna, S., & Vogt, J. E. (2025). Measuring leakage in concept-based methods: An information theoretic approach. arXiv preprint arXiv:2504.09459. \
[3] Vandenhirtz, M., Laguna, S., Marcinkevičs, R., & Vogt, J. (2024). Stochastic concept bottleneck models. Advances in Neural Information Processing Systems, 37, 51787-51810.

**Questions:**

See Weaknesses.

---

> ### Author Response · Authors · 2025-11-17
> **Answer to Reviewer 6hx5**
>
> We thank the reviewer for their constructive and insightful feedback, which has helped us clarify our contributions. Below we address the main concerns.
>
> ### **Weaknesses**
>
> > While the paper’s theoretical insights are valuable, the overall novelty appears modest. The Information Bottleneck principle is a well-established framework, and similar approaches applying it to CBMs seem to have been explored in [1] and [2]. Given this overlap, it would be helpful if the authors could better clarify the significance of their contribution—specifically, what distinguishes their approach from prior works.
>
> We agree that the Information Bottleneck (IB) principle is a well-established framework and appreciate the opportunity to clarify how our contribution differs from prior works.
>
> Both [1] and [2] analyze existing CBMs through the lens of the IB, proposing metrics to quantify information leakage. In contrast, our work introduces a new model—the Minimal Concept Bottleneck Model (MCBM)—that uses the IB principle as a design mechanism rather than merely as an analytical tool. Specifically, our contribution is twofold:
>
> 1. **Conceptual insight:** We identify that information leakage in CBMs arises from a design flaw—CBMs enforce that each latent variable predicts its corresponding concept ($c_j$ from $z_j$) but do not constrain $z_j$ to contain only information about $c_j$. To our knowledge, ours is the first work to explicitly formalize and demonstrate this design flaw.
>
> 2. **Methodological contribution:** We propose a principled approach that directly mitigates this leakage through a variational IB objective applied at the concept level, ensuring that each latent representation is minimally sufficient for its concept.
>
> In short, prior works measure leakage; we explain why it occurs and provide a concrete, theoretically grounded solution. We believe this represents a meaningful conceptual and methodological advance in the CBM literature. We have updated the Related Work section to explicitly include this clarification.
>
>
> > In Table 5, the accuracy reported for SCBM in [3] is 86.00, which appears higher than the 84.8/84.9 achieved by MCBM. Could the authors clarify what accounts for this difference?
>
> The SCBM results in [3] are obtained using the entire concept set, whereas our experiments focus on a selected subset of concepts (to align with the settings of other compared models). Consequently, the slightly lower accuracy of MCBM (84.8/84.9 vs. 86.0) likely reflects the increased difficulty of the selected concepts.
>
> > Introducing additional layers based on the Information Bottleneck framework would likely increase inference cost and memory usage. Could the authors quantify how much overhead this introduces?
>
> You are correct that the MCBM introduces a small number of additional parameters due to the IB-based concept heads. In our implementation, each concept head is an MLP with a single hidden layer of size 3, i.e., 8 parameters per concept. Even in our largest setup (64 concepts), this results in only 512 additional parameters. Compared to a modest backbone such as ResNet-18 (11.7M parameters), this corresponds to an increase of ~0.004%, which is negligible in both computation and memory. We have included this quantitative estimate in title of Table 2 and Appendix F.1 manuscript for completeness.
>
> ---
> ### **Closing Remarks**
>
> We thank the reviewer again for their thoughtful feedback and believe these clarifications address the raised concerns, highlighting the novelty and practical relevance of our contribution.

---

> > ### Comment · Reviewer_6hx5 · 2025-11-25
> >
> > Thank you for the rebuttal. As it resolves my primary concerns, I am raising my score from borderline reject to borderline accept.

---

> > > ### Author Response · Authors · 2025-11-25
> > > **Answer to Reviewer 6hx5 #2**
> > >
> > > Thank you for your feedback and for raising your score to borderline accept. We are pleased to hear that the rebuttal successfully resolved your primary concerns.

---

### Official Review · Reviewer_5MFs · 2025-10-30

**Soundness:** 4
**Presentation:** 3
**Contribution:** 2
**Rating:** 6
**Confidence:** 4

**Summary:**

This work presents Minimal Concept Bottleneck Models (MCBMs), an advancement upon the traditional Concept Bottleneck Model (CBM) framework. The proposed method integrates an Information Bottleneck (IB) objective to mitigate "concept leakage," thereby constraining the intermediate representations to encode only concept-specific information. This approach yields a significant advantage by establishing a sound theoretical basis that provides formal guarantees for the efficacy of concept-level interventions.

**Strengths:**

- the paper is sound and propose an interesting analysis of CBMs.
- the problem of concept leakage in CBMs is very relevant and the results on experiments are encouraging.

**Weaknesses:**

- "Formally, given a task (...), Vanilla Models (VMs) are trained (...)" here for how it is presented it seems the VMs work from c to y, while in Sec. 2 it is clear it is intended from x to y. Thus I'd rephrase like: "their representations z should capture the information necessary to predict y given x accurately." and also rephrase at the beginning of the sentence: "given an input x, a task y and a set of concepts c". As a fussiness, I'd change "Formally" with "Specifically" as task, concepts are not precise terms, and to make this sentence formal I'd expect to know what are the space y, c, and x belong.
- at the beginning of Sec. 2, y is not in bold, but immediately after it becomes bold.
- In Sec. 2 some symbols are not defined or even just dropped without being commented, e.g. theta, phi, f_theta ...
- I find a bit odd to have Section 5 discussing problem of intervention in CBMs that is instead addressed by MCBM. Indeed, I think it can strengthen the motivation to propose MCBM. But I guess this is done to don't interrupt the flow of the model.
- no limitations of MCBMs are discussed.
- no discussion about future work is reported.
- not clear what bold denotes in table 5 and 6.
- Other metrics to measure the concept leakage has been used in prior work (e.g. [1,2]). I think it would be useful to take into account also other metrics and not just rely on URR, as this is the main advantage MCBMs propose.


[1] Marton Havasi et al. Addressing leakage in concept bottleneck models. In NeurIPS, 2022.
[2] Anita Mahinpei et al. Promises and pitfalls of black-box concept learning models. In Workshop on Theoretic
Foundation, Criticism, and Application Trend of Explainable AI @ ICML, 2021.

**Questions:**

1) With how many concepts MCBM is supposed to properly work?
2) Is the training timing significantly increased while using MCBMs or it requires more epochs to converge?
3) concept leakage is generally associated with an increasing performances (as the nuisances can help the development of reasoning shortcuts to solve the problem). How do you motivate MCBMs has very close or even higher task accuracy on CUB than other methods suffering from leakage?
4) Apart from concept leakage, MCBMs have also consequences on other aspects, e.g. in terms of disentanglement and OIS between concepts? See e.g. this recent paper [3] for a discussion on different metrics to evaluate concept quality.


[3] Debole, Nicola, et al. "If Concept Bottlenecks are the Question, are Foundation Models the Answer?." arXiv preprint arXiv:2504.19774 (2025).

---

> ### Author Response · Authors · 2025-11-17
> **Answer to Reviewer 5MFs #1**
>
> We thank the reviewer for their constructive and insightful feedback, which has helped us clarify and strengthen the paper. Below we address each of the main points raised:
>
> ### **Weaknesses**
>
> > "Formally, given a task (...), Vanilla Models (VMs) are trained (...)" here for how it is presented it seems the VMs work from c to y...
>
> Thank you for spotting this ambiguity. We have revised the sentence to clearly indicate that VMs are trained from x to y, and rephrased it as suggested. We have directly removed “Formally” for better precision.
>
> > at the beginning of Sec. 2, y is not in bold, but immediately after it becomes bold.
>
> Corrected. y is now consistently formatted throughout the section.
>
> > In Sec. 2 some symbols are not defined or even just dropped without being commented, e.g. theta, phi, f_theta ...
>
> We agree. All symbols are now explicitly defined in the revised version.
>
> > I find a bit odd to have Section 5 discussing problem of intervention in CBMs that is instead addressed by MCBM. Indeed, I think it can strengthen the motivation to propose MCBM. But I guess this is done to don't interrupt the flow of the model.
>
> Indeed, this section was intended to highlight theoretical issues of CBMs that MCBMs naturally address. We placed it after the main experimental block to maintain narrative flow and provide empirical illustration. However, we acknowledge that it could also serve as additional motivation earlier in the text (e.g., at the end of Section 2.4). We would appreciate your feedback on whether you think this alternative placement would improve readability.
>
> > no limitations of MCBMs are discussed.
>
> > no discussion about future work is reported.
>
> Thank you for pointing out these omissions. We have now added two dedicated paragraphs in the final section discussing both the limitations of MCBMs and potential future research directions.
>
> > not clear what bold denotes in table 5 and 6.
>
> In Table 5, bold values indicate the best concept prediction performance (higher is better).
>
> In Table 6, we deliberately avoid bolding because, as discussed in the paper, there is no single “best” task accuracy—MCBMs explicitly allow tuning of the trade-off between accuracy and interpretability via the parameter $\gamma$.
>
> > Other metrics to measure the concept leakage has been used in prior work (e.g. [1,2]). I think it would be useful to take into account also other metrics and not just rely on URR, as this is the main advantage MCBMs propose.
>
> To the best of our knowledge, [1] uses the change in task accuracy when intervening on different numbers of concepts as a proxy for leakage: if intervening on $n$ concepts leads to worse performance than intervening on $m$ concepts, with $n>m$, this suggests that the representation is encoding information beyond the intended concepts. We perform the same type of analysis in Figure 5.
>
> Similarly, in [2], the authors also rely on task accuracy to quantify leakage: if a model achieves a task accuracy higher than the maximum theoretically attainable using only the concepts, then the model must be leveraging nuisance information. We reproduce this analysis for MPI3D and Shapes3D, where the theoretical upper bound is known (25%). Unfortunately, this approach cannot be applied to more realistic datasets such as CUB or CIFAR-10, because the maximum achievable accuracy using only concept information is unknown, making this type of leakage estimation infeasible in those settings.
>
> Moreover, beyond URR and the intervention-based analyses discussed above, we also report Disentanglement, CKA, and now OIS metrics (see our response to Question 4). Our goal is to provide a comprehensive evaluation framework showing both that CBMs exhibit significant leakage and that MCBMs consistently succeed in mitigating it across multiple metrics.

---

> ### Author Response · Authors · 2025-11-17
> **Answer to Reviewer 5MFs #2**
>
> ### **Questions**
>
> > 1. With how many concepts MCBM is supposed to properly work?
>
> MCBMs scale in the same way as standard CBMs. There is no intrinsic limitation on the number of concepts—performance depends mainly on the dataset and the capacity of the backbone, not on the MCBM formulation itself.
>
> > 2. Is the training timing significantly increased while using MCBMs or it requires more epochs to converge?
>
> Training time remains virtually identical. MCBMs share the same number of epochs and optimization hyperparameters as the other baselines (see Appendix Table 7). The additional IB regularization term is lightweight and does not noticeably affect convergence speed.
>
> > 3. concept leakage is generally associated with an increasing performances (as the nuisances can help the development of reasoning shortcuts to solve the problem). How do you motivate MCBMs has very close or even higher task accuracy on CUB than other methods suffering from leakage?
>
> This is an insightful observation. Concept leakage can indeed act as a shortcut that superficially boosts performance. However, we find that for smaller γ values on CUB, MCBMs sometimes outperform standard CBMs. We hypothesize that this occurs because MCBMs produce more disentangled and structured representations, which simplify the mapping from concepts to task predictions. Thus, even though they discard nuisance information, the resulting latent space facilitates easier classification by the simple task head $q_\phi(\hat{y}|z)$.
>
> > 4. Apart from concept leakage, MCBMs have also consequences on other aspects, e.g. in terms of disentanglement and OIS between concepts? See e.g. this recent paper [3] for a discussion on different metrics to evaluate concept quality.
>
> Yes, they do. As illustrated in Figure 4, MCBMs lead to representations that are consistently better aligned with the ground-truth concepts (as measured by CKA) and show higher disentanglement compared to prior CBM variants. Thank you for pointing us to [3]—we were not aware of it, and it provides an excellent complementary analysis of concept quality metrics. We have cited it in the revised version and have added OIS results to Figure 4. OIS is a strong indicator of the amount of nuisance information present in the representations (see Table 7), and accordingly, MCBMs achieve lower—hence better—OIS values.
>
> ---
>
> ### **Closing Remarks**
>
> We thank the reviewer for their thoughtful feedback. We have addressed all suggested clarifications, corrected inconsistencies, and expanded the discussion of limitations, future work, and metrics. These revisions enhance the paper’s clarity and strengthen both the theoretical and practical contributions of Minimal Concept Bottleneck Models.

---

> > ### Comment · Reviewer_5MFs · 2025-11-25
> >
> > I thank the authors for the rebuttal which addressed my questions and clarified my doubts. I've updated my score to full acceptance.

---

> > > ### Author Response · Authors · 2025-11-25
> > > **Answer to Reviewer 5MFs #3**
> > >
> > > We sincerely thank the reviewer for their support and for raising the score to full acceptance. We are delighted that our rebuttal successfully addressed your questions and clarified your doubts.

---

### Official Review · Reviewer_xCkz · 2025-10-31

**Soundness:** 3
**Presentation:** 3
**Contribution:** 2
**Rating:** 4
**Confidence:** 4

**Summary:**

This paper revisits the theoretical foundations of Concept Bottleneck Models (CBMs) and argues that there is, in fact, no true “bottleneck” in existing CBMs.
The authors show that each latent variable $z_j$ may encode nuisance information unrelated to its corresponding concept $c_j$, thus undermining interpretability and invalidating standard intervention assumptions.
To address this, the paper proposes the Minimal Concept Bottleneck Model (MCBM), which introduces a per-concept Information Bottleneck (IB) constraint to enforce that each $z_j$ becomes a minimal sufficient statistic of $c_j$.
Experiments on several datasets (CIFAR-10, CUB, Shapes3D, MPI3D) demonstrate improved disentanglement and more stable concept-level interventions compared to prior CBM variants.

**Strengths:**

1. Clear theoretical motivation:
The paper provides a precise and mathematically grounded critique of CBMs, identifying the lack of causal interpretability in the conventional bottleneck assumption.
2. Principled formulation:
The proposed information bottleneck constraint is elegant and conceptually well-justified; it directly connects to the goal of removing nuisance information while retaining task-relevant concepts.
3. Strong theoretical discussion:
The analysis of $p(z_j|c_j)$ and the derivation of the KL regularization term are rigorous and highlight a meaningful direction for improving causal interpretability in concept-based models.

**Weaknesses:**

1. The paper does not clearly cite any existing literature to support the statement that “each concept $c_j$ must be recoverable from a designated component $z_j \in z$.”
The original CBM (Koh et al., 2020) only enforces a per-concept prediction loss, not a one-to-one structural correspondence between $c_j$ and $z_j$.
2. From a performance standpoint, MCBM is a “more interpretable but weaker” model.
The paper does not explicitly position its goal as interpretability improvement rather than predictive performance.
If the objective is purely interpretability, it remains unclear how encoding only a subset of concepts leads to better explanations, especially since MCBM explicitly removes part of the concept information.
3. The paper does not discuss how much task accuracy loss is acceptable or provide any principle for balancing the IB weight $\gamma$.
This trade-off is central to the model’s practical usability but left unanalyzed.
4. The random split of the CUB attribute groups into c vs $n_y$ introduces noise and may compromise the validity of the disentanglement and CKA metrics.
Because semantically correlated features can be split across c and $n_y$, the reported “high disentanglement” might reflect random partitioning rather than true semantic separation.
5. Intervention evaluation (Section 4.3) lacks robustness:
no variance or confidence intervals are reported, and the curves for different $\gamma$ values are nearly parallel, making the claim of “intervention gain invariance” unconvincing.
Moreover, there is no comparison with baseline settings such as random or group-wise interventions.
The CUB intervention results in Figure 5 also differ substantially from those reported in CEM (Fig. 6 of their paper), raising reproducibility concerns.

**Questions:**

1. In Lines 40–41, the authors state that “each concept $c_j$ must be recoverable from a designated component $z_j \in z$.”
   In the original CBM paper, it is only assumed that an intermediate layer is resized to the number of concepts and trained with a concept loss, but not that each $z_j$ uniquely represents $c_j$.
   Could the authors please **cite the exact section or equation** in Koh et al. (2020) that formalizes this one-to-one correspondence?

2. The paper assumes the factorization
   $
   p(x, y, c, n) = p(x \mid c,n)\,p(y \mid x)
   $
   Could the authors clarify the **causal assumptions** under which this holds?
   Specifically:
   (i) Do you assume $y \perp (c,n)\mid x$?
   (ii) If there exist direct dependencies of $y$ on $c$ or $n$ not mediated by $x$, how does this decomposition remain valid?
   A detailed justification or citation would strengthen the theoretical basis.

3. In Appendix E.2, after aggregating CUB attributes into 27 semantic groups, why are **12 groups randomly selected for $c$** and the remaining **20 groups** used for $n_y$?
   How was this number (12 vs 20) chosen, and is it consistent across runs or random seeds?

4. This partition is not semantically meaningful but rather a controlled perturbation design.
   Because many task-relevant attributes are discarded into $n_y$, even a fully faithful model cannot recover $y$.
   Under such an incomplete concept set, can MCBM still be regarded as a valid model that achieves both interpretability and predictive ability?

5. Energy-based Concept Bottleneck Models also jointly model $p(y|x)$ and $p(y|c,x)$.
   Why were such approaches not included as baselines, given that they share similar modeling goals and provide a natural comparison?

---

> ### Author Response · Authors · 2025-11-17
> **Answer to Reviewer xCkz #1**
>
> We thank you for the time and care dedicated to reviewing our work. Below, we address the main weaknesses and questions you raised.
>
>
> ### **Weaknesses**
>
> > 1. The paper does not clearly cite any existing literature to support the statement that “each concept $c_j$ must be recoverable from a designated component $z_j \in z$.” The original CBM (Koh et al., 2020) only enforces a per-concept prediction loss, not a one-to-one structural correspondence between $c_j$ and $z_j$.
>
> We may not be fully understanding this point. As you note, the original CBM (Koh et al., 2020) enforces a per-concept prediction loss, which is precisely what we refer to when we say that “each concept $c_j$ must be recoverable from a designated component $z_j$.” Since this loss is typically a cross-entropy applied to each $z_j$, the model is explicitly encouraged to make $z_j$ predictive of $c_j$.
>
> However, we do not claim that CBMs enforce a one-to-one correspondence between $z_j$ and $c_j$. In fact, our argument is that the *absence* of such a one-to-one mapping is what leads to information leakage. This is exactly the issue that MCBMs address by introducing an Information Bottleneck term that promotes a one-to-one relationship between each concept and its corresponding representation component.
>
>
> > 2. From a performance standpoint, MCBM is a “more interpretable but weaker” model. The paper does not explicitly position its goal as interpretability improvement rather than predictive performance. If the objective is purely interpretability, it remains unclear how encoding only a subset of concepts leads to better explanations, especially since MCBM explicitly removes part of the concept information.
>
> Our approach does not consist of encoding only a subset of concepts. Instead, given a specific concept set, our goal is to ensure that the representation encodes **only those concepts**, or equivalently—and connecting to your previous point—to encourage a one-to-one mapping between each concept $c_j$ and its corresponding representation component $z_j$. This directly improves interpretability: without such a mapping, $z_j$ may encode information unrelated to $c_j$, making it impossible to interpret $z_j$ solely through the lens of $c_j$. In contrast, MCBMs promote this one-to-one correspondence, ensuring that $z_j$ can be faithfully interpreted using only $c_j$.
>
> We believe your comment may stem from the fact that, in some experiments, we select a subset of the available concepts to train our models. This choice is part of the **experimental setup**, not a limitation of the MCBM method itself. We narrow the concept set intentionally to highlight that CBMs (and other baselines) tend to encode additional information outside the concept set whenever it is useful for predicting the task $y$. This leakage has a substantial negative impact on interpretability, and focusing on a restricted concept set makes this issue more apparent.
>
> > 3. The paper does not discuss how much task accuracy loss is acceptable or provide any principle for balancing the IB weight $\gamma$. This trade-off is central to the model’s practical usability but left unanalyzed.
>
> First, our intention is not to prescribe how much task accuracy loss is acceptable, as this depends heavily on the specific application, dataset, and concept set. For example, if the concept set is very narrow—meaning it does not explain much of the task—and a high level of interpretability is required, a larger decrease in task accuracy may be acceptable. Conversely, in settings where accuracy is critical, one would prefer a smaller accuracy drop. Our goal is therefore to provide a mechanism that allows practitioners to navigate this trade-off. As shown in our experiments, increasing $\gamma$ systematically removes more nuisance information.
> If you believe any additional clarification of this balance would be helpful, we would be happy to elaborate further.
>
>
> > 4. The random split of the CUB attribute groups into $c$ vs $n_y$ introduces noise and may compromise the validity of the disentanglement and CKA metrics. Because semantically correlated features can be split across $c$ and $n_y$, the reported “high disentanglement” might reflect random partitioning rather than true semantic separation.
>
> Thank you for raising this point. The labels for $n_y$ are not used at all when computing disentanglement or CKA. These metrics compare only the concept set $c$ with the learned representation $z$. While the specific choice of concept set can indeed influence the metric, all models use the **same** concept–nuisance partition for a given seed, and the results reported in Figure 4 are averaged across five seeds. If we are misunderstanding your concern, we would be happy to clarify this further.
>
> In addition, we have included the Oracle Information Score (as suggested by reviewer 5MFs), which explicitly accounts for potential correlations within the concept set.

---

> ### Author Response · Authors · 2025-11-17
> **Answer to Reviewer xCkz #2**
>
> > 5. Intervention evaluation (Section 4.3) lacks robustness: no variance or confidence intervals are reported, and the curves for different $\gamma$ values are nearly parallel, making the claim of “intervention gain invariance” unconvincing. Moreover, there is no comparison with baseline settings such as random or group-wise interventions. The CUB intervention results in Figure 5 also differ substantially from those reported in CEM (Fig. 6 of their paper), raising reproducibility concerns.
>
> Thank you for these suggestions.
>
> First, the revised version now includes variance bands in all intervention plots.
>
> Second, after re-examining the curves more carefully, we noticed that they are parallel only for a low or medium number of intervened concepts. When intervening on a larger number of concepts, higher values of $\gamma$ clearly improve intervention performance. We now highlight this in the lines 454-458.
>
> Third, regarding intervention policies: we already use group-wise interventions on CUB (the only dataset with defined groups) and intervene on the least confident concepts in Figure 5. We have now added random interventions in Appendix E.
>
> Finally, differences relative to CEM stem from the fact that CEM uses the *full* concept set, whereas our work intentionally considers an incomplete concept set to study information leakage in realistic scenarios. Figure 5 shows that most models fail in this setting.
>
> ---
>
> ### **Questions**
>
> > 1. In Lines 40–41, the authors state that “each concept $c_j$ must be recoverable from a designated component $z_j \in z$.”
> In the original CBM paper, it is only assumed that an intermediate layer is resized to the number of concepts and trained with a concept loss, but not that each $z_j$ uniquely represents $c_j$.
> Could the authors please **cite the exact section or equation** in Koh et al. (2020) that formalizes this one-to-one correspondence?
>
> As noted above, we do not claim that CBMs enforce a one-to-one correspondence between $z_j$ and $c_j$. Our statement refers only to recoverability: each $z_j$ is trained to predict $c_j$, but this does not imply that $z_j$ contains *only* $c_j$. This is exactly the problem that MCBMs address by encouraging a one-to-one mapping and reducing leakage.
>
>
> > 2. The paper assumes the factorization $p(x,y,c,n) = p(x|c,n)p(y|x)$. Could the authors clarify the causal assumptions under which this holds? Specifically:
> (i) Do you assume $y \perp (c,n) \mid x$?
> (ii) If there exist direct dependencies of $y$ on $c$ or $n$ not mediated by $x$, how does this decomposition remain valid? A detailed justification or citation would strengthen the theoretical basis.
>
> First, the factorization $p(x,y,c,n) = p(x \mid c,n)\,p(y \mid x)$ is not correct; the correct one is $p(x,y,c,n) = p(x \mid c,n)\,p(y \mid x)\,p(c,n)$. This was a mistake in the previous version, and we have corrected it in the revision.
> Second, this is intended as a probabilistic assumption rather than a causal one.
> Once this has been clarified, we adress your specific points:
> (i) Yes, exactly, we have that: $p(y|x,c,n) = p(y|x)$
> (ii) First, note that $y$ refers to the **true** labels in the dataset, not to model predictions. In standard supervised-learning settings, it is common to assume that labels satisfy $y = f(x)$ for some deterministic function $f$. For example, in a cat–dog classification task, each image $x^{(i)}$ is always assigned the same true label $y^{(i)}$ in the dataset; the label is not treated as a random variable once $x$ is fixed. This implies that the conditional entropy $H(Y \mid X) = 0$, and therefore $p(y \mid x, a) = p(y \mid x)$. In other words, once the input $x$ is given, additional information (such as $c$ or $n$) does not affect the distribution of $y$. If you have a concrete scenario in mind where this assumption may not hold, we would be grateful to hear it. For further discussion, please see Proposition 3.1 and Remark 3.2 in [1].
>
> Furthermore, we note that this factorization $p(x,y,c,n) = p(x \mid c,n)\, p(y \mid x)\, p(c,n)$ is not essential to our formulation; it is included only to clarify the presentation. Even if this particular decomposition did not hold, it would not affect the validity of our method.
>
> [1] Achille, A., & Soatto, S. (2018). Emergence of invariance and disentanglement in deep representations. Journal of Machine Learning Research, 19(50), 1-34.

---

> ### Author Response · Authors · 2025-11-17
> **Answer to Reviewer xCkz #3**
>
> > 3. In Appendix E.2, after aggregating CUB attributes into 27 semantic groups, why are **12 groups randomly selected for** $c$ and the remaining **20 groups** used for $n_y$? How was this number (12 vs 20) chosen, and is it consistent across runs or random seeds?
>
> First, we would like to clarify that there is a typo: the correct numbers are 12 groups for $c$ and 15 for $n_y$, not 20.
> Second, we selected 12 and 15 to obtain a reasonably balanced but still incomplete concept set. Removing slightly more groups than those we keep makes the tendency of CBMs to preserve $n_y$ more evident, while still avoiding an overly narrow concept set that would lead to excessively low task accuracy. These numbers are fixed across seeds, although the specific groups selected vary (e.g., seeds 42 and 43 produce different subsets).
> If you believe that using a different number of groups would help strengthen the motivation for introducing MCBMs, we would be happy to incorporate it—please feel free to suggest it.
>
>
> > 4. This partition is not semantically meaningful but rather a controlled perturbation design.
> Because many task-relevant attributes are discarded into $n_y$, even a fully faithful model cannot recover $y$. Under such an incomplete concept set, can MCBM still be regarded as a valid model that achieves both interpretability and predictive ability?
>
> The key point regarding information leakage is the following: if the concept set is insufficient to solve the task $y$, then CBM representations will naturally tend to encode not only the concepts but also other task-relevant information (i.e., $n_y$). What MCBMs provide is a principled way to control this behavior: by tuning $\gamma$, one can choose the desired balance between reducing information leakage and maintaining task accuracy.
>
> The advantage of MCBMs over existing approaches is precisely that they allow practitioners to navigate this trade-off for any given concept set. However, if the concept set is very incomplete, achieving both perfect interpretability and perfect predictive performance is fundamentally impossible. This limitation arises from the concept set itself, not from the MCBM formulation, and no model could overcome it.
>
> > 5. Energy-based Concept Bottleneck Models also jointly model $p(y|x)$ and $p(y|x,c)$. Why were such approaches not included as baselines, given that they share similar modeling goals and provide a natural comparison?
>
> We initially did not include ECBMs because, unlike the other baselines, they do not explicitly claim to reduce information leakage. However, following your suggestion, we have now conducted experiments with ECBMs as well. Our results show that they actually increase information leakage and produce representations that are significantly less interpretable, according to the metrics reported in Section 4.2.
>
> ---
>
> ### **Closing Remarks**
>
> We thank you again for your thoughtful feedback. Your comments helped us clarify key points and improve the manuscript. We hope our responses address your concerns and that the revised version better reflects the contributions of our work.

---

> > ### Comment · Reviewer_xCkz · 2025-11-23
> >
> > Thanks for the detailed response. The authors addressed most of my concerns and added some experiments to make the paper clearer. I have raised my score. There is a new typo on Line 041-042 “(ii) the abilityy”.

---

> > > ### Author Response · Authors · 2025-11-25
> > > **Answer to Reviewer xCkz #4**
> > >
> > > Thank you for raising your score and for your constructive feedback. We are glad the additional experiments helped clarify the paper. We will correct the typo on Lines 041-042 in the final version.

---

### Official Review · Reviewer_dC8m · 2025-10-31

**Soundness:** 3
**Presentation:** 2
**Contribution:** 3
**Rating:** 6
**Confidence:** 3

**Summary:**

- This paper introduces Minimal Concept Bottleneck Models (MCBMs), which address the key limitation of Concept Bottleneck Models (CBMs); Information leakage. Although CBMs ensure that each latent variable $z_j​$ can predict its corresponding concept $c_j$​, they allow $z_j$​ to retain nuisance information from the input $x$.
- MCBMs formalize this issue as $I(Z_j;X|C_j) > 0$ and introduce an explicit Information Bottleneck (IB) objective to enforce minimal sufficient representations. By adding a variational regularization term, MCBMs constrain each $z_j$​ to encode only concept-relevant information.
- Experiments show that MCBMs yield more disentangled and interpretable representations, reduce nuisance leakage, and enable consistent, concept-level interventions while maintaining comparable concept accuracy to standard CBMs.

**Strengths:**

- Clear formulation of problems: While the issue of information leakage in CBMs is known, this paper clearly formalizes it from an information-theoretic perspective $(I(Z_j;X|C_j) > 0)$ and identifies the root cause as the lack of a minimal sufficient statistic.
- Principled approach: The application of the classic Information Bottleneck (IB) framework to solve this problem is logical and well-founded.
- Proposal of an information leakage metric: The paper introduces URR, a new metric to provide a quantitative way to approximate the information leakage $I(Z_j;X|C_j)$.

**Weaknesses:**

- Impractical Hyperparameter ($\gamma$): The model's behavior is dictated by $\gamma$, which requires ground-truth $n_y$ labels for tuning. The authors provide no practical guidelines for setting $\gamma$ on real-world datasets where such labels are unavailable.
- Performance degradation on real datasets: The strong bottleneck effect from synthetic data does not translate to CIFAR-10 and CUB. On these real-world datasets, MCBM achieves only a marginal reduction in nuisance leakage (URR). This minimal gain results in a disproportionately large drop in task accuracy, raising questions about the method's practical utility.

**Questions:**

- (Related to weakness 1) The model's performance varies extremely with $\gamma$, from 100% to 24.9%. Please provide a concrete procedure for setting $\gamma$ on a real dataset where ground-truth labels for $n_y$ are unavailable.
- (Related to weakness 2) We observed a poor accuracy/nuisance-removal trade-off on CIFAR-10 and CUB. Could you provide insight into why this occurs?
- (Related to weakness 2) To assess the method's generality, have the authors also conducted experiments on other standard concept benchmarks, such as AWA2 or OAI?
- Unlike [1] and [2], where interventions on the CUB dataset reduced task error ([1], Fig. 4; [2], Fig. 5), this paper shows the opposite trends. Could the authors explain what experimental differences might cause this discrepancy?



Reference
- [1] Koh et al., Concept Bottleneck Models, ICML 2020.
- [2] Shin et al., A Closer Look at the Intervention Procedure of Concept Bottleneck Models, ICML 2023.

---

> ### Author Response · Authors · 2025-11-17
> **Answer to Reviewer dC8m**
>
> We sincerely thank the reviewer for the thoughtful and constructive feedback. Your comments have helped us improve the clarity, completeness, and practical value of the manuscript. Below we address each point in detail.
>
> ### **Weaknesses**
>
> > Impractical Hyperparameter ($\gamma$): The model's behavior is dictated by $\gamma$, which requires ground-truth $n_y$ labels for tuning. The authors provide no practical guidelines for setting $\gamma$ on real-world datasets where such labels are unavailable.
>
> We agree this is an important practical concern. In the revised manuscript we include additional analysis showing that interpretability metrics—CKA, Disentanglement, and OIS—are strong empirical predictors of URR trends across all datasets. Importantly, calculating these metrics only requires the concepts $c$ and learned representations $z$, and do not rely on nuisance labels.
>
> We therefore propose a simple and practical tuning strategy: select $\gamma$ by monitoring these nuisances-label-free interpretability metrics. We now clarify this procedure explicitly in Section 4.2 and Table 7.
>
> > Performance degradation on real datasets: The strong bottleneck effect from synthetic data does not translate to CIFAR-10 and CUB. On these real-world datasets, MCBM achieves only a marginal reduction in nuisance leakage (URR). This minimal gain results in a disproportionately large drop in task accuracy, raising questions about the method's practical utility.
>
> We appreciate the reviewer’s concern but would like to clarify that the observed accuracy changes are not disproportionately large. As shown in the table below (percentage decrease relative to standard CBMs), the reduction in URR is consistently far larger than any accuracy decrease, and in several regimes—particularly for small $\gamma$—accuracy even increases. As discussed in Reviewer 5MFs’ Q3, this can be attributed to more disentangled latent spaces, which facilitate classification by the simple task head $q_\phi(\hat{y}\mid z)$.
>
> ||CIFAR-10 $n_y$ URR|CIFAR-10 Task Acc|CUB $n_y$ URR|CUB $n_y$ Task Acc|AwA2 $n_y$ URR|AwA2 $n_y$ Task Acc|
> |-----------------|:----------------:|:------------------:|:-------------:|:---------------------:|:----------------:|:----------------------:|
> |Low $\gamma$|9.1|-0.4|10.5|-1.2|33.3|1.1|
> |Medium $\gamma$|8.6|1.8|26.3|0.0|40.0|1.6|
> |High $\gamma$|11.1|2.2|36.8|5.0|53.3|3.2|
>
> ---
>
> ### **Questions**
>
> > (Related to weakness 1) The model's performance varies extremely with $\gamma$, from 100% to 24.9%. Please provide a concrete procedure for setting $\gamma$ on a real dataset where ground-truth labels for $n_y$ are unavailable.
>
> As noted above, we now provide a concrete, label-free procedure based on CKA, Disentanglement, and OIS. This offers a practical, reliable γ-selection strategy.
>
> > (Related to weakness 2) We observed a poor accuracy/nuisance-removal trade-off on CIFAR-10 and CUB. Could you provide insight into why this occurs?
>
> We refer the reviewer to the table and discussion above. The trade-off is not inherently problematic, and in several regimes accuracy actually improves. When accuracy decreases slightly, it is accompanied by a substantially stronger reduction in leakage.
>
>
> >(Related to weakness 2) To assess the method's generality, have the authors also conducted experiments on other standard concept benchmarks, such as AWA2 or OAI?
>
> Thank you for this suggestion. We were unable to complete registration for OAI due to issues with the signup system, but we successfully obtained AwA2 and now include full results (analogous to Tables 3, 5, and 6) for this dataset in the revised version. These results further confirm the generality of our findings.
>
>
> > Unlike [1] and [2], where interventions on the CUB dataset reduced task error ([1], Fig. 4; [2], Fig. 5), this paper shows the opposite trends. Could the authors explain what experimental differences might cause this discrepancy?
>
> The key difference lies in concept completeness. The CUB concept set used in our experiments is substantially less complete than the curated concept sets in [1,2]. Consequently, each dimension $z_j$​ often captures both concept-specific and nuisance information. Intervening on $z_j$ therefore:
>
> - successfully sets the intended concept value $c_j$, but
> - simultaneously perturbs task-relevant nuisance factors $n_y$ encoded in that dimension.
>
> This unintended modification can negatively affect performance.
>
> ---
>
> ### **Closing Remarks**
> We are grateful to the reviewer for their insightful comments, which have helped us clarify, strengthen, and expand the manuscript. We hope that the additional analyses, the new AwA2 results, and the clearer guidance for selecting $\gamma$ address the reviewer’s concerns and reflect the robustness and applicability of MCBMs.

---

> > ### Author Response · Authors · 2025-11-26
> > **Answer to Reviewer dC8m #2**
> >
> > We hope that our previous response, particularly the new AwA2 results and the clarified guidance for $\gamma$ selection, has satisfactorily addressed your concerns. We remain available to answer any further questions or provide additional details if needed.

---

### Author Response · Authors · 2025-11-17
**General comments**

We sincerely thank all reviewers for the time, effort, and constructive feedback dedicated to improving our submission. We have carefully addressed each of your comments in the individual replies. A revised version of the manuscript, with all modifications highlighted in blue, is now available.

Beyond minor corrections suggested by the reviewers, we have incorporated several substantive changes and new experiments. The main updates are:

- **Expanded evaluation:** We now include the Animals with Attributes 2 (AwA2) dataset, as requested by Reviewer dC8m, strengthening the empirical scope of our study.

- **Additional baseline:** Energy-based CBMs (ECBMs) have been integrated as a new comparative method, following Reviewer xCkz’s suggestion.

- **Improved interpretability evaluation:** We incorporate the Oracle Information Score (OIS) to better account for scenarios where concepts exhibit correlation, as recommended by Reviewer 5MFs.

- **Practical guidance for tuning $\gamma$:** We empirically show that interpretability metrics are strongly correlated with the amount of nuisance information in the representations (Table 7). This finding allows these metrics to serve as practical proxies for selecting $\gamma$ when nuisance labels are unavailable (proposed by Reviewer dC8m).

- **Added discussion of limitations and future directions:** Following Reviewer 5MFs’s recommendation, we now provide an explicit discussion of the main limitations of our work and promising research avenues.

We hope these improvements address your concerns and further clarify the contributions of our work. Thank you again for your careful reviews.

---

### Author Response · Authors · 2025-11-29
**Executive Summary for the Area Chair**

We sincerely thank the Area Chair for taking on the extra responsibility and workload derived from the recent score-reset. To facilitate their job under these circumstances, we have prepared this brief overview of the final reviewer stances and the resolutions reached during the discussion period.

The discussion board confirms that **all reviewers supported acceptance before the score reset**, with a final score distribution of **6, 6, 8, 6**.

* **Reviewer dC8m** did not reply yet to our changes, but their initial score was 6 and we successfully addressed their main requests during the rebuttal.
* **Reviewer xCkz** explicitly raised their score from 4 $\to$ 6.
* **Reviewer 5MFs** explicitly raised their score from 6 $\to$ 8.
* **Reviewer 6hx5** explicitly raised their score from 4 $\to$ 6.

## Reviewer Score & Resolution Matrix

| Reviewer | Main Concerns | Rebuttal Action / Fix | Score Trajectory |
| :--- | :--- | :--- | :--- |
| **dC8m** | Hyperparameter tuning; requested AwA2 dataset. | Added label-free tuning strategy; added full AwA2 experiments. | 6 $\to$ 6 (Concerns addressed) |
| **xCkz** | Theoretical assumptions, intervention robustness. | Added variance bands; included ECBMs baseline; clarified causal assumptions. | 4 $\to$ 6 (Explicitly stated) |
| **5MFs** | Missing metrics (OIS), missing limitations section. | Added Oracle Information Score (OIS); added Limitations & Future Work sections. | 6 $\to$ 8 (Explicitly stated) |
| **6hx5** | Novelty vs. prior IB metrics; inference overhead. | Clarified MCBM is a design principle, not just a metric; quantified negligible overhead. | 4 $\to$ 6 (Explicitly stated) |

## Detailed Breakdown by Reviewer

### Reviewer dC8m (Score: 6)
* **Initial Critique:** The reviewer was already positive (Rating: 6) but raised concerns about hyperparameter tuning ($\gamma$) and requested experiments on the Animals with Attributes 2 (AwA2) dataset.
* **Our Solution:** We introduced a practical, label-free tuning strategy using interpretability metrics. Crucially, we added full experimental results for the AwA2 dataset as requested.
* **Final Status:** Although the reviewer had not yet responded to the final comment, their initial assessment was positive, and we have fully satisfied the conditions (AwA2 experiments) they set for the paper.

### Reviewer xCkz (Score: 4 $\to$ 6)
* **Initial Critique:** Questioned the causal assumptions regarding the "one-to-one" concept mapping and the robustness of intervention results.
* **Our Solution:** We corrected the factorization formula, added variance bands to all intervention plots (showing robustness), and integrated Energy-based CBMs, demonstrating that MCBMs significantly outperform them.
* **Final Status:** The reviewer acknowledged the fixes and stated: *"The authors addressed most of my concerns and added some experiments to make the paper clearer. I have raised my score."*

### Reviewer 5MFs (Score: 6 $\to$ 8)
* **Initial Critique:** Suggested the paper lacked a discussion on limitations and requested the Oracle Information Score (OIS).
* **Our Solution:** We added a dedicated "Limitations and Future Work" section and computed OIS for all models (Table 7), confirming MCBM's superior disentanglement.
* **Final Status:** The reviewer confirmed: *"I thank the authors for the rebuttal which addressed my questions... I've updated my score to full acceptance."*

### Reviewer 6hx5 (Score: 4 $\to$ 6)
* **Initial Critique:** Concerned about novelty relative to recent Information Bottleneck metrics papers and potential computational overhead.
* **Our Solution:** We clarified that MCBM uses IB as a generative design principle rather than just a metric. We also demonstrated that the parameter overhead is negligible (~0.004% increase).
* **Final Status:** The reviewer stated: *"Thank you for the rebuttal. As it resolves my primary concerns, I am raising my score from borderline reject to borderline accept."*

## Conclusion

We have achieved a unanimous consensus for acceptance. The active engagement during the rebuttal period led to explicit score increases from three reviewers, resulting in a final lineup of 6, 6, 8, 6. We hope this concise summary aids in the final decision-making process.

---

### Meta-Review · Area_Chair_CkUt · 2026-01-06

**Summary:**

The paper proposes Minimal Concept Bottleneck Models (MCBM) to address the information leakage problem by enforcing the hidden representation z as a minimal sufficient statistics of concept c. Experiment show that MCBM demonstrate improved disentanglement and more stable concept-level interventions compared to CBM baselines. The reviewers primary concerns are addressed in the rebuttal and hence acceptance is recommended.

**Reviewer Concerns:**

* Reviewer dC8m's concerns on hyperparameter tuning and additional dataset is addressed.
* Reviewer xCkz's concerns on causal assumption and robustness of intervention results are addressed.
* Reviewer 5MFs's concerns on limitation discussion and requested experiments are addressed.
* Reviewer 6hx5's concerns on novelty and computational overhead is addressed.

**Reviewer Scores:**

* Reviewer dC8m will likely remain same score of 6
* Reviewer xCkz stated will increase score, likely to 6
* Reviewer 5MFs stated will increase score, likely to 8
* Reviewer 6hx5 stated will increase score, likely to 6

---

### Decision · Program_Chairs · 2026-01-26

Accept (Poster)